# DNA-mediated association of two histone-bound complexes of yeast Chromatin Assembly Factor-1 (CAF-1) drives tetrasome assembly in the wake of DNA replication

Francesca Mattiroli[1]*[†], Yajie Gu[1,2][†], Tejas Yadav[3,4], Jeremy L Balsbaugh[5], Michael R Harris[4], Eileen S Findlay[1], Yang Liu[1], Catherine A Radebaugh[2], Laurie A Stargell[2,6], Natalie G Ahn[7], Iestyn Whitehouse[4], Karolin Luger[1,6]*

[1]Department of Chemistry and Biochemistry, Howard Hughes Medical Institute, University of Colorado Boulder, Boulder, United States; [2]Department of Biochemistry and Molecular Biology, Colorado State University, Fort Collins, United States; [3]Weill Cornell Graduate School of Medical Sciences, New York, United States; [4]Molecular Biology Program, Memorial Sloan Kettering Cancer Center, New York, United States; [5]Department of Chemistry and Biochemistry, University of Colorado Boulder, Boulder, United States; [6]Institute for Genome Architecture and Function, Colorado State University, Fort Collins, United States; [7]Biofrontiers Institute, University of Colorado Boulder, Boulder, United States

*For correspondence: francesca. mattiroli@colorado.edu (FM); karolin.luger@colorado.edu (KL)

[†]These authors contributed equally to this work

Competing interests: The authors declare that no competing interests exist.

**Abstract** Nucleosome assembly in the wake of DNA replication is a key process that regulates cell identity and survival. Chromatin assembly factor 1 (CAF-1) is a H3-H4 histone chaperone that associates with the replisome and orchestrates chromatin assembly following DNA synthesis. Little is known about the mechanism and structure of this key complex. Here we investigate the CAF-1•H3-H4 binding mode and the mechanism of nucleosome assembly. We show that yeast CAF-1 binding to a H3-H4 dimer activates the Cac1 winged helix domain interaction with DNA. This drives the formation of a transient CAF-1•histone•DNA intermediate containing two CAF-1 complexes, each associated with one H3-H4 dimer. Here, the (H3-H4)$_2$ tetramer is formed and deposited onto DNA. Our work elucidates the molecular mechanism for histone deposition by CAF-1, a reaction that has remained elusive for other histone chaperones, and it advances our understanding of how nucleosomes and their epigenetic information are maintained through DNA replication.

## Introduction

Dynamic assembly and disassembly of nucleosomes regulates accessibility to the genome during the processes of DNA transcription, replication and repair. DNA replication constitutes a particularly challenging context, as nucleosomes and the epigenetic information they encode need to be reestablished and duplicated onto daughter strands (*Alabert and Groth, 2012*). This process involves a number of histone chaperones that operate through a network of sequential interactions (*Groth et al., 2007*; *Huang et al., 2015*; *Jasencakova et al., 2010*; *Richet et al., 2015*; *Saredi et al., 2016*; *Smith and Stillman, 1989*; *Tyler et al., 1999*). Histone chaperones bind histones (*Elsasser and D'Arcy, 2013*), and are responsible for the maintenance of nucleosome density and the faithful inheritance of the epigenetic information.

While knowledge of how histone chaperones bind histones is steadily increasing (*Huang et al., 2015*; *Mattiroli et al., 2015*; *Richet et al., 2015*; *Ricketts et al., 2015*; *Saredi et al., 2016*), the mechanism and structural transitions required for histone deposition onto DNA remain unknown. This is a non-trivial process in light of the complex architecture of the histone octamer in the nucleosome, requiring the ordered deposition of histone pairs. This is particularly relevant for the first step of nucleosome assembly, the deposition of the $(H3-H4)_2$ tetramer (tetrasome formation).

Parental and newly-synthesized H3-H4 transition between a dimeric and a tetrameric state, while replication-coupled histone chaperones distribute them onto the daughter strands during DNA replication (*Campos et al., 2010*; *Clement and Almouzni, 2015*; *English et al., 2006*; *Huang et al., 2015*; *Richet et al., 2015*). Current evidence favors a conservative model for H3-H4 inheritance, where parental $(H3-H4)_2$ tetramers are not split and distributed onto the daughter strands but rather deposited as one unit (*Xu et al., 2010*), but alternative models have also been proposed (*Tagami et al., 2004*).

CAF-1 is the key nucleosome assembly factor associated with DNA replication (*Smith and Stillman, 1989*). CAF-1 directly interacts with the replisome via the processivity factor PCNA (*Krawitz et al., 2002*; *Moggs et al., 2000*; *Rolef Ben-Shahar et al., 2009*; *Shibahara and Stillman, 1999*; *Zhang et al., 2000*) and bridges interactions with epigenetic factors (*Loyola et al., 2009*; *Murzina et al., 1999*; *Quivy et al., 2008*). The unique role of CAF-1 in integrating chromatin assembly with DNA synthesis and epigenetic signaling makes it indispensable for the maintenance of cell identity and for life in multicellular organisms (*Barbieri et al., 2014*; *Cheloufi et al., 2015*; *Houlard et al., 2006*; *Ishiuchi et al., 2015*; *Nakano et al., 2011*; *Song et al., 2007*). In yeast, CAF-1 deletion is viable but results in aberrant transcriptional silencing programs and sensitivity to DNA damage (*Kaufman et al., 1998*).

CAF-1 is a histone H3-H4 chaperone, composed of three distinct subunits, all of which are conserved from yeast (named Cac1, Cac2 and Cac3) to humans (*Almouzni and Méchali, 1988*; *Kaufman et al., 1995*, *1997*; *Smith and Stillman, 1989*; *Tyler et al., 1999*; *Verreault et al., 1996*). Previous biochemical studies have suggested that one CAF-1 complex can bind more than one H3-H4 dimer (*Liu et al., 2012*), and other studies have proposed models where CAF-1 dimerization may be important for its function in vivo (*Nakano et al., 2011*; *Quivy et al., 2001*). To date, the mechanism by which CAF-1, and indeed any histone chaperone, assembles the $(H3-H4)_2$ tetramer onto DNA is unknown. Understanding how CAF-1 deposits H3-H4 is essential for our understanding of the mechanisms that govern the inheritance of epigenetic modifications and epigenome maintenance.

Here, we describe the mechanism by which CAF-1-mediates deposition of the $(H3-H4)_2$ tetramer onto DNA. A coordinated sequence of events is set in motion by H3-H4 binding to CAF-1, and promoted by CAF-1•DNA interaction. The mechanism culminates in the DNA-mediated association of two CAF-1•H3-H4 complexes to form the $(H3-H4)_2$ tetramer, which is then transferred to DNA. These findings elucidate the histone deposition mechanism by this key histone chaperone complex, with significant implications on our understanding of chromatin propagation in DNA replication.

## Results

### tCAF-1 is sufficient for nucleosome assembly in vitro

Whether recombinant CAF-1 assembles nucleosomes in vitro in absence of the replication machinery or other nuclear proteins remains an open question (*Gaillard et al., 1996*; *Kadyrova et al., 2013*; *Kaufman et al., 1995*; *Smith and Stillman, 1991*), which we set out to resolve in a purified in vitro system. We developed a quantitative nucleosome assembly assay (NAQ). Commonly used nucleosome assembly assays rely on native gel-based readouts, where the formation of the canonical nucleosome band is monitored over a range of histone chaperone concentrations. This readout is not suitable for proteins that bind DNA or the histone•DNA products, such as FL CAF-1 (*Figure 1a*, lanes 3–6, and *Figure 1—figure supplement 1a*). Therefore, we added a Micrococcal Nuclease (MNase) digestion step that allows us to monitor the DNA protection pattern induced by histone deposition. A further DNA purification step and addition of a DNA loading control allows accurate size determination and quantification of the recovered DNA fragments, hence the formation of nucleosomes.

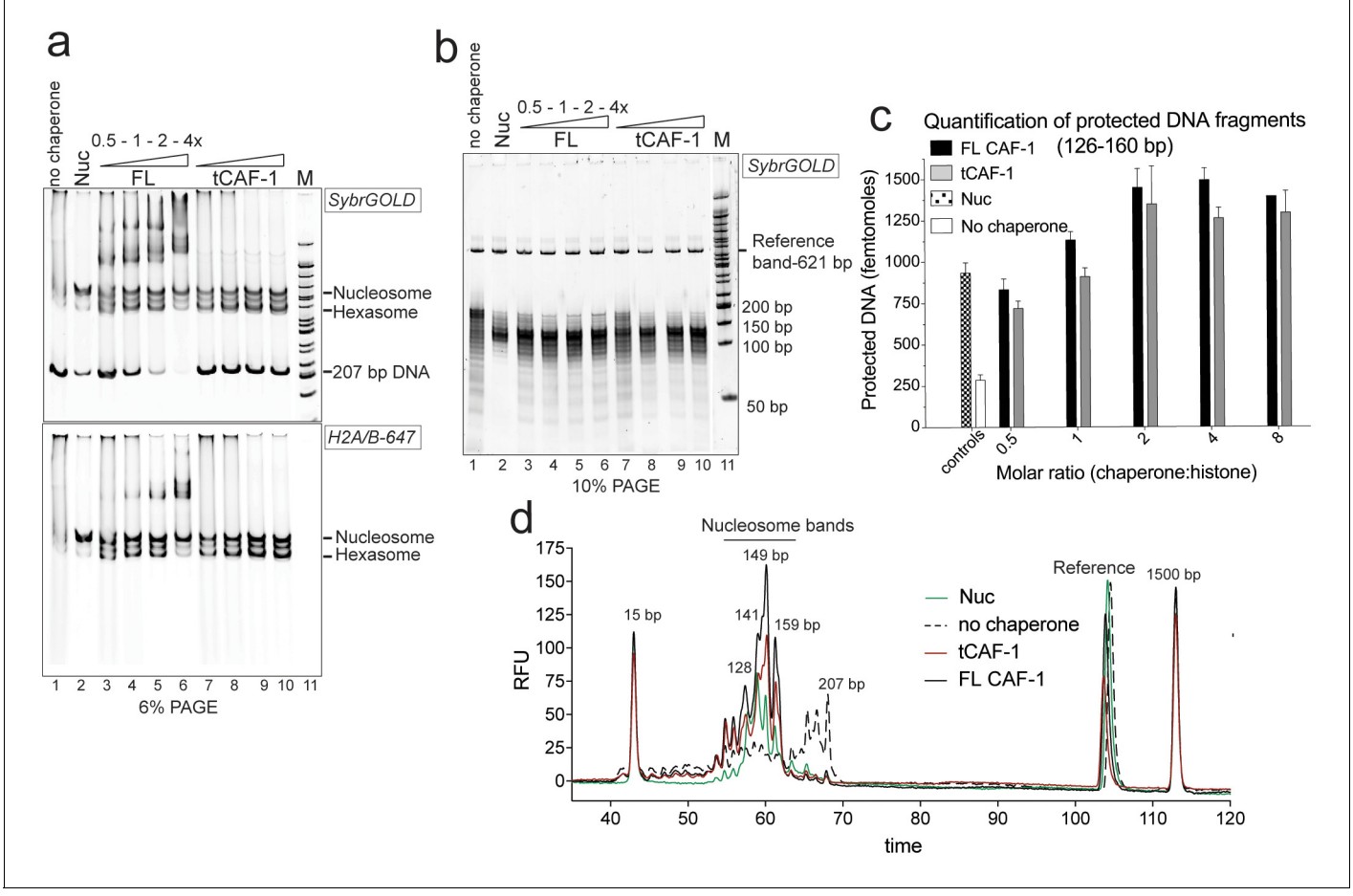

**Figure 1.** tCAF-1 is competent for nucleosome assembly in absence of other factors. (**a**) Products from the nucleosome assembly assay on 207 bp DNA, with FL CAF-1 or tCAF-1. DNA is 200 nM, (H3–H4)$_2$ 200 nM and H2A-H2B-AttoN647 400 nM; 100-200-400-800 nM CAF-1 is titrated. Assemblies were analyzed by native PAGE. (**b**) Products of MNase digestion performed on samples shown in panel **a**, after DNA purification. The 207 bp band in lane 1 results from nonproductive association of histones with DNA that renders it MNase-resistant. The disappearance of the 207 bp protection as CAF-1 is titrated demonstrates the ability of CAF-1 to relieve non-specific histone-DNA complexes, as observed for other chaperones. (**c**) Quantification of protected nucleosomal DNA (126–160 bp) obtained from the samples shown in panel **a** and **b**. The reference DNA is used to normalize amounts in each lane. Mean ± SD is shown for at least three repeats. The data used for this panel is included in *Figure 1—source data 1*. (**d**) Bioanalyzer electropherograms of protected DNA fragments from nucleosome assembly with 400 nM FL CAF-1 and tCAF-1 shown in **b**, with controls from salt-assembled nucleosomes (Nuc) and a no-chaperone sample.

The following source data and figure supplement are available for figure 1:

**Source data 1.** FL and tCAF-1 assemble nucleosomes in vitro.

**Figure supplement 1.** Control experiments for *Figure 1*.

We used the NAQ assay to study the in vitro nucleosome assembly activity of FL CAF-1 and of a truncated tCAF-1 complex lacking the first 233 amino acids in Cac1, but containing all the determinants for CAF-1 subunit assembly and histone binding (*Mattiroli et al., 2017*). The nucleosome assembly reactions on 207 bp DNA (*Figure 1a*) suggested efficient nucleosome formation by both FL and tCAF-1. Notably, tCAF-1 doesn't bind DNA or its products (*Figure 1a*, lanes 7–10), suggesting that the N-terminal portion of Cac1 mediates DNA binding observed with FL CAF-1. The assembly reactions were subjected to MNase digestion, and the purified DNA was analyzed by native PAGE (*Figure 1b*) and quantified using a Bioanalyzer (Agilent) (*Figure 1c–d*). DNA fragments of about 125–160 bp accumulated with both CAF-1 constructs, similar to what was obtained with salt-

assembled nucleosomes (Nuc, *Figure 1b–d*). To validate that these DNA fragments are representative of nucleosomes and not of other DNA•protein complexes, we performed a number of control reactions, including analysis of the CAF-1•DNA complexes, none of which result in significant DNA protection under these conditions (*Figure 1—figure supplement 1b*, box).

These experiments demonstrate that tCAF-1 and FL CAF-1 are both active and efficient nucleosome assembly factors in absence of other nuclear proteins. Our results also indicate that the N-terminal portion of Cac1, which in vivo is associated with PCNA, is not required for the nucleosome assembly activity, and that tCAF-1 contains all the requisite components for histone deposition. Nucleosome assembly activity increases in a CAF-1 dose dependent manner, and reaches its maximum at a two-fold excess of CAF-1 complex per (H3-H4)$_2$ tetramer (*Figure 1c*). No protected bands were detected above or below the nucleosomal DNA fragments (*Figure 1—figure supplement 1c*), confirming that the primary product of CAF-1 are nucleosomes and hexasomes. Because H2A-H2B can spontaneously associate with tetrasomes in vitro (*Figure 1—figure supplement 1d*), and because CAF-1 itself has significantly lower affinity for H2A-H2B compared to H3-H4 (*Figure 1—figure supplement 1e*), it appears that the primary role of CAF-1 is to promote the formation of an ordered (H3-H4)$_2$•DNA complex, the tetrasome, as noted previously (*Smith and Stillman, 1991*).

## CAF-1 has one binding site for a H3-H4 dimer

To understand the mechanism of CAF-1 mediated nucleosome assembly, we first investigated how it binds histones. On CAF-1, the H3-H4 binding site is formed by the acidic region of Cac1 (*Liu et al., 2016*), in conjunction with Cac1-bound Cac2 (*Mattiroli et al., 2017*). Published data suggest that CAF-1 facilitates the formation of a (H3-H4)$_2$ tetramer (*Liu et al., 2012*, *2016*). H3-H4 exists in an equilibrium between dimeric and tetrameric states in absence of DNA or a histone chaperone. The estimated dissociation constant for tetramerization of the H3-H4 dimer is considerably weaker than the affinity of H3-H4 for DNA or for the histone chaperones (*Donham et al., 2011*). We therefore tested whether CAF-1 binds H3-H4 in its dimeric or tetrameric form. We used sedimentation velocity analytical ultracentrifugation (SV-AUC) to characterize the complexes formed between CAF-1 and histone H3-H4 combined at different molar ratios (1:1 or 1:2 CAF-1 to H3-H4 dimer), comparing wild type H3-H4 (WTH3-H4) with a constitutively dimeric H3-H4 mutant (DMH3-H4) that binds CAF-1 with the same affinity as WTH3-H4 (*Figure 2—figure supplement 1a–b*). Adding a 1:1 ratio of WTH3-H4 or DMH3-H4 dimer to CAF-1 resulted in the same homogenous shifts in S$_{20,w}$ value, seen both with FL and tCAF-1 (*Figure 2a* and *Figure 2—figure supplement 1c*). Adding two WTH3-H4 dimers per CAF-1 molecule results in further apparent increase in size, possibly representing binding of a WT(H3-H4)$_2$ to one CAF-1. On the contrary, with DMH3-H4, which is unable to form a tetramer, we observed no additional increase (*Figure 2a* and *Figure 2—figure supplement 1c*). These experiments were performed at 4 μM concentration where WTH3-H4 is expected to be primarily tetrameric, while DMH3-H4 remains a dimer. These data suggest that CAF-1 has a single binding site for H3-H4 and that this interaction may not affect tetramerization of (H3-H4)$_2$, as a shift to higher S$_{20,w}$ value is not observed with a dimeric DMH3-H4.

To further test this hypothesis, we used FRET-based Job plot experiments. Here, we used wild-type histones, constitutively dimeric DMH3-H4 and constitutively tetrameric XL(H3-H4)$_2$ prepared by cross-linking two H3-H4 dimers via a single cysteine residue on H3K115C (*D'Arcy et al., 2013*). This reaction yields a XL(H3-H4)$_2$ preparation where about 80% of the histones are tetramers (*Figure 2—figure supplement 1d*), in a conformation that is compatible with the (H3-H4)$_2$ observed in the nucleosome structure. Notably, XL(H3-H4)$_2$ binds to FL or tCAF-1 with the same affinity as WTH3-H4 (*Figure 2—figure supplement 1e*). In stoichiometry experiments, performed under conditions where WTH3-H4 is primarily in its dimeric form, we observe one CAF-1 complex binding to one WT or DMH3-H4 dimer, while two CAF-1 complexes are bound to one XL(H3-H4)$_2$ tetramer. This is observed for both FL and tCAF-1 (*Figure 2b* and *Figure 2—figure supplement 1f*), supporting the idea that each CAF-1 complex has a single binding site for one H3-H4 dimer and confirming that the Cac1 N-terminus has no role in H3-H4 binding. Further confirmation for this stoichiometry comes from SEC-MALS (size exclusion chromatography coupled to multi-angle light scattering) experiments with tCAF-1, where significant increases in apparent molar mass are observed with XL(H3-H4)$_2$ but not with WTH3-H4, compatible with two tCAF-1 complexes binding to the XL(H3-H4)$_2$ tetramer (*Figure 2c* and *Figure 2—figure supplement 1g*).

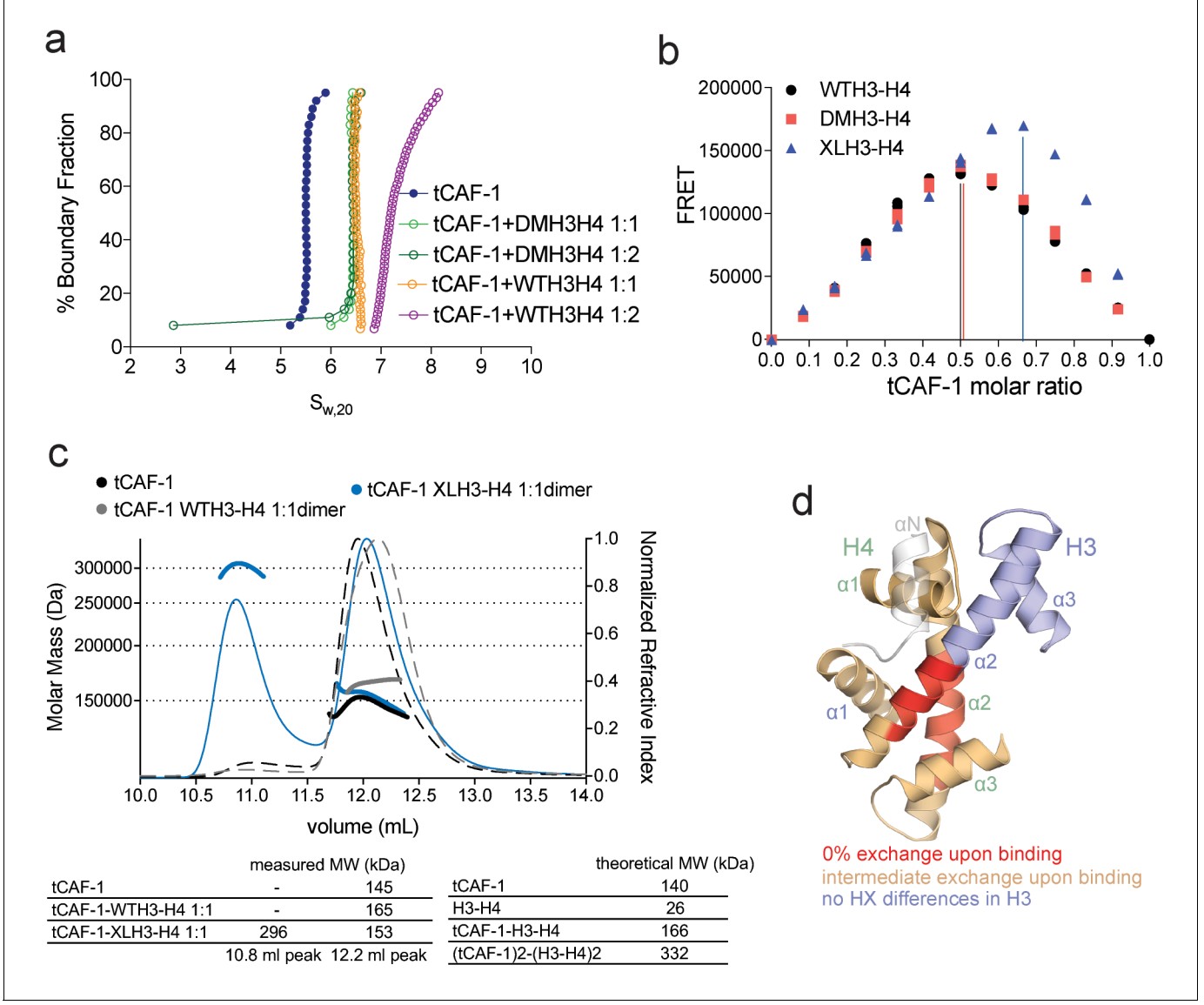

**Figure 2.** CAF-1 has a single binding site for a H3-H4 dimer. (**a**) van Holde-Weischet analysis of SV-AUC runs performed with tCAF-1 and WTH3-H4, titrated in a 1:1 or 1:2 molar ratio of CAF-1 to H3-H4 dimer (orange and purple). A constitutively dimeric form of H3-H4, DMH3-H4 (green) was analyzed likewise. (**b**) FRET-based Job plot assay with tCAF-1 and H3-H4 shows a single binding event with WT or DMH3-H4 dimers, while with XL(H3-H4)$_2$ a stoichiometry of two CAF-1 complexes per tetramer is observed (total protein concentration was kept at 150 nM, where WT H3-H4 is dimeric). Two independent measurements are depicted; the data points are mostly overlapping. (**c**) SEC-MALS experiment of tCAF-1 alone or in complex with WT or XL(H3-H4)$_2$. The sample containing XL(H3-H4)$_2$ shows an additional peak at ~300 kDa. The protein elution traces (refractive index, RI) refer to the right y axis, the calculated molar masses refer to the left y axis. Validation that tCAF-1 binds WTH3-H4 on SEC is shown in *Figure 2—figure supplement 1g*. (**d**) Protection of regions on the H3-H4 dimer (from PDB: 1AOI) upon interaction with tCAF-1, based on HX-MS experiments (*Figure 2—figure supplement 2*). Red areas: regions with complete protection on both WT and DMH3 upon interaction with tCAF-1 (near 0% uptake upon binding); orange regions: sites of intermediate HX protection upon interaction with tCAF-1. Blue indicate s regions with no significant protection in H3, upon binding to tCAF-1. H3 αN is displayed with transparency because it is likely in a different conformation in the free H3-H4 dimer; no peptide coverage was observed for this region.

The following figure supplements are available for figure 2:

**Figure supplement 1.** Control experiments for *Figure 2*.

**Figure supplement 2.** HX-MS data on CAF-1•H3-H4 complexes.

Altogether, these experiments indicate that each CAF-1 complex has a single binding site for one H3-H4 dimer, and that CAF-bound H3-H4 is still able to tetramerize, either with or without its own bound CAF-1 complex (*Figure 2—figure supplement 1h*). These assemblies are not stable enough to be isolated using WTH3-H4, but are observed with XL(H3-H4)$_2$. This suggests that the four-helix bundle formed by two H3 molecules in the (H3-H4)$_2$ tetramer is important for this complex formation and perhaps the only point of contact between the two CAF-1 moieties.

To further characterize the tCAF-1•H3-H4 complex, we used hydrogen-deuterium exchange coupled to mass spectrometry (HX-MS). CAF-1 binding to H3-H4 induces protection from exchange of an extended area on both WT and DMH3-H4, as opposed to a distinct region of protection. This suggests that CAF-1 binding to histones elicits a global conformational effect, which was also observed for other chaperone-histone complexes (*D'Arcy et al., 2013*; *DeNizio et al., 2014*) (*Figure 2d* and *Figure 2—figure supplement 2a*). The most significant changes are clustered at the α1-α2 region of both histones (*Figure 2d* and *Figure 2—figure supplement 2a*). This global conformational effect makes it difficult to identify the direct binding interfaces on H3-H4 (*Figure 2—figure supplement 2b*). Importantly, no significant changes in HX were detected in the H3 region mediating H3-H4 tetramerization through a four-helix bundle (α3; aa 111–126) (*Figure 2d* and *Figure 2—figure supplement 2a*), confirming our interpretation that this region is exposed and available for tetramer formation in the CAF-1•H3-H4 complex. Together, these data indicate that CAF-1 binds to one H3-H4 dimer that results in its overall stabilization, and that permits tetramerization in its CAF-1 bound form.

## H3-H4 binding activates DNA binding by the Cac1 winged helix domain (WHD)

In our HX-MS experiments we observed HX changes distributed throughout all three CAF-1 subunits upon H3-H4 binding, with no direct evidence pointing towards a specific interface (*Figure 2—figure supplement 2c*). The observed HX changes are in agreement with recently published data from similar HX-MS experiments using the FL CAF-1 complex (*Liu et al., 2016*), confirming that deleting the N-terminal portion of Cac1 has no significant effect on histone binding. Moreover, by using both WT and DMH3-H4, we further support the idea that the mutations in DMH3-H4 do not affect its interaction with CAF-1 (*Figure 2—figure supplement 2c*). The broadly distributed HX changes suggest extensive structural rearrangements in the CAF-1 complex upon histone binding, with likely implications for the mechanism of histone deposition. In particular, we noticed that the C-terminal Cac1 WHD, recently identified as a DNA binding domain (*Zhang et al., 2016*), showed deprotection upon histone binding in the HX-MS analysis (*Figure 2—figure supplement 2c*), also seen when FL CAF-1 was used (*Liu et al., 2016*). Interestingly, tCAF-1 does not bind DNA in absence of histones, despite containing a functional WHD domain (*Figure 1—figure supplement 1b* and below).

We therefore asked if an intramolecular interaction between the histone binding region and the WHD may be masking the DNA binding activity of tCAF-1 in absence of histones. The acidic region, which constitutes part of the histone binding interface (*Mattiroli et al., 2017*; *Liu et al., 2016*), is a likely candidate for mediating this intramolecular interaction due to its complementary charges to the basic WHD. Indeed, mutant tCAF-1 complexes in which the Cac1 acidic region was neutralized or deleted, namely tCAF_Nac and tCAF_Δac respectively, gained DNA binding activity (*Figure 3a–b*, lanes 5–10). This indicates that in absence of histones the WHD may be inhibited by an intramolecular interaction with the acidic region on Cac1, a component of the histone binding module (*Liu et al., 2016*) together with Cac2 (*Mattiroli et al., 2017*) (histone binding module is referred to as HBM, *Figure 3a* and *Figure 3—figure supplement 1a*). We therefore performed a competition experiment by adding the HBM of CAF-1, or only Cac2 as control, to DNA-bound isolated WHD. Strikingly, the HBM could compete the WHD away from DNA, while Cac2 alone could not (*Figure 3c*). Moreover, using in solution cross-linking experiments we were able to trap the complex between the WHD and the HBM, a complex that contains the acidic region and Cac2, but not between the WHD and Cac2 (*Figure 3—figure supplement 1b*). Together, these data suggest that the Cac1 acidic domain engages in an inhibitory intramolecular interaction with the WHD, thereby masking its DNA binding capacity.

To directly test if H3-H4 binding can unmask the DNA binding activity of the WHD, we established a gel-based DNA binding assay with tCAF-1 and histones. Here, pre-mixed stoichiometric tCAF-1•H3-H4 complexes were combined with limiting amounts of DNA and then analyzed on native

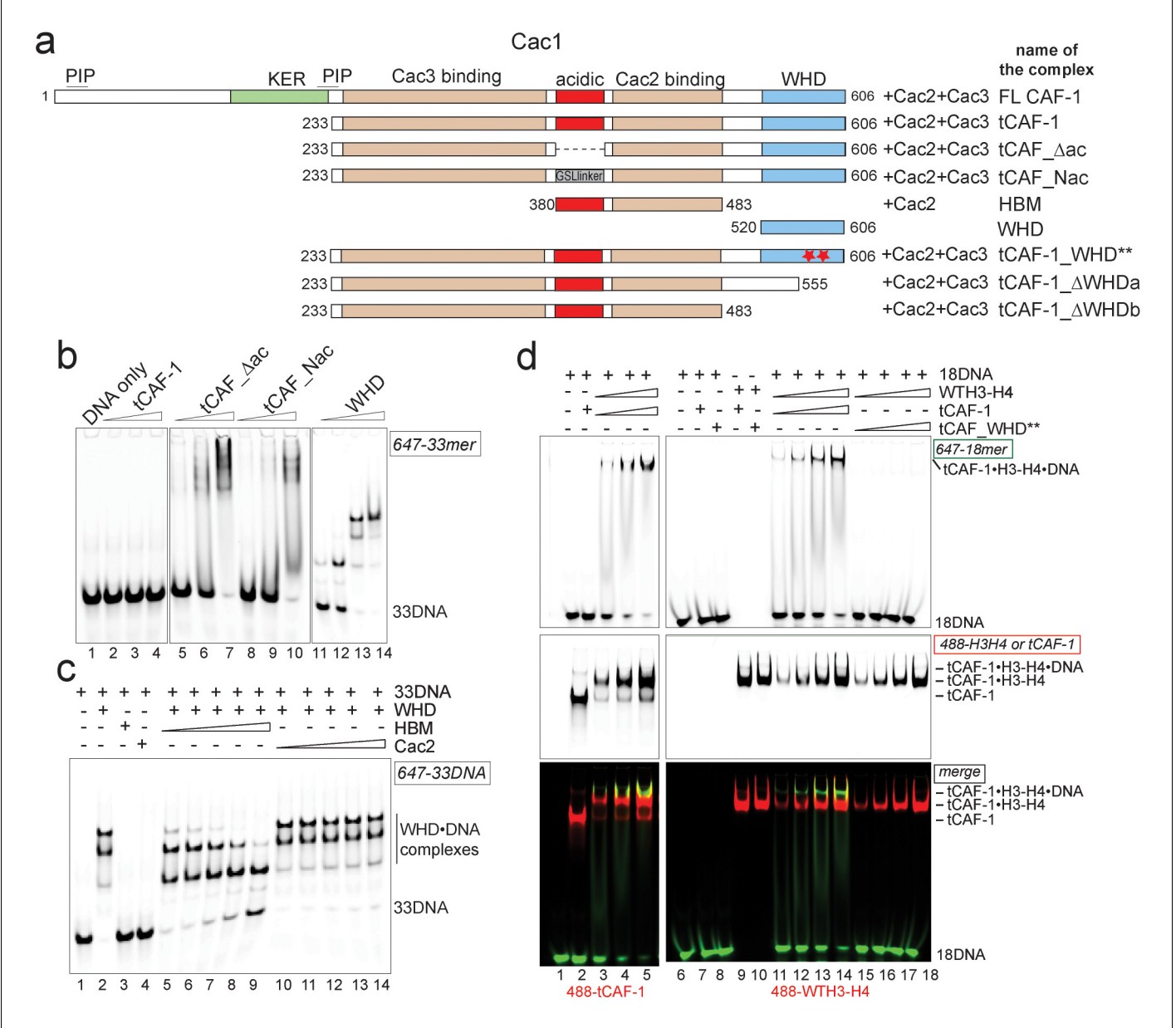

**Figure 3.** H3-H4 binding activates DNA binding by the CAF-1 complex. (a) Schematic of the constructs and complexes used in this and subsequent figures. Cac1 domains that are combined with full length Cac2 and/or Cac3 as indicated, are depicted. PIP stands for PCNA binding peptide; tCAF_WHD** contains mutations at K564E and K568E. tCAF_Δac contains a deletion of residues 397–431 in Cac1, while in tCAF_Nac aa 397–431 were replaced with a Gly-Ser-Leu linker. KER identifies a Lys-Arg-Glu rich region. The Cac2 and Cac3 binding regions were mapped using HX-MS. Gel filtration profiles and SDS PAGE of these complexes are shown in *Figure 3—figure supplement 1a*. (b) The acidic domain of Cac1 inhibits DNA binding by tCAF-1. 100 nM 33 bp DNA (33DNA) was mixed with tCAF-1, isolated WHD, or the tCAF-1 complexes mutated in the acidic region (depicted in a). CAF-1 complexes are titrated as 1–0.5-0.25–0.125 μM. (c) The acidic domain competes WHD away from DNA. The WHD was kept constant at 1 μM and DNA at 100 nM. Cac2 and HBM were titrated as 4-2-1-0.5–0.25 μM. (d) Histone binding releases the DNA binding activity of tCAF-1. EMSA performed with 100 nM 18 bp DNA (18DNA). tCAF-1 is titrated 0.37–0.75-1.5–3 μM, and was added either alone (in control lanes) or as a 1:1 complex with WTH3-H4 (H3-H4 dimer concentration) to the DNA. Alexa488-labeled tCAF-1 was used in lanes 1–5, while Alexa488-labeled H3-H4 were used in lanes 6–18.

The following figure supplement is available for figure 3:

**Figure supplement 1.** Control experiments for *Figure 3*.

gels to evaluate the formation of ternary tCAF-1•H3-H4•DNA complexes. We used an 18 bp DNA (18DNA) fragment, which represents the minimal substrate for binding to the WHD domain (*Gajiwala et al., 2000*), but is of insufficient length for wrapping the $(H3-H4)_2$ tetramer. This is important, as using longer DNA promotes rapid histone deposition and would therefore not allow us to demonstrate the DNA binding activity of histone-bound tCAF-1. As expected, tCAF-1 did not bind to 18DNA in absence of histones (*Figure 3d*, lane 2). However, when tCAF-1 was pre-incubated with histones, we observed a distinct high molecular weight complex, migrating slower than the tCAF-1•H3-H4 complex without DNA, and containing DNA, H3-H4, and tCAF-1 (*Figure 3d*, lanes 3–5 and 11–14). To confirm that DNA binding in this ternary tCAF-1•H3-H4•18DNA complex is mediated through the Cac1 WHD, we tested tCAF-1 complexes with a mutated WHD (tCAF_WHD**, containing Cac1 K564E K568E, *Figure 3a*). This mutant complex does not form the DNA-bound intermediate (*Figure 3d*, lanes 15–18), even though it still binds H3-H4 (*Figure 3d*, lane 10) with the same affinity and stoichiometry as tCAF-1 (*Figure 3—figure supplement 1c–e*). These data strongly suggest that H3-H4 triggers DNA binding by the WHD, by releasing it from an inhibitory intramolecular interaction with the histone binding region.

## DNA promotes the association of two CAF-1 complexes to form the $(H3-H4)_2$ tetramer

This ternary complex may well represent a relevant intermediate in the histone deposition mechanism of CAF-1. To determine the structural organization of the tCAF-1•H3-H4•18DNA complex, we investigated the effect of using a constitutively dimeric DMH3-H4 or a constitutively tetrameric XL $(H3-H4)_2$ in this context. First, we wondered if a dimeric DMH3-H4 could form this ternary intermediate. Indeed, tCAF-1•DMH3-H4 assembles a similar complex with 18DNA to what observed with tCAF-1•WTH3-H4 (*Figure 4a*, lanes 11–14) indicating that histone tetramerization is not required for the formation of this intermediate. This in turn also supports the idea that CAF-1 binds an H3-H4 dimer, and that binding to a histone H3-H4 dimer is sufficient to release the WHD and engage it in DNA binding. In addition, we wondered whether a constitutively tetrameric XL$(H3-H4)_2$ was still capable to assemble this intermediate. As shown in *Figure 4b*, tCAF-1•XL$(H3-H4)_2$ interacts with 18DNA forming the ternary tCAF-1•XL$(H3-H4)_2$•18DNA complex, hence XL$(H3-H4)_2$ binding is also able to displace the Cac1 WHD.

Because we have reported that two tCAF-1 complexes can bind to XL$(H3-H4)_2$ in absence of DNA (*Figure 2b* and *Figure 2—figure supplement 1f*), we wanted to test whether this stoichiometry was maintained after DNA binding. To this end, we used tCAF-1 mutated in the WHD, in complex with XL$(H3-H4)_2$ (*Figure 4c*). We saw no interaction of (tCAF_WHD**)$_2$•XL$(H3-H4)_2$ with 18DNA, confirming that the Cac1 WHD is the primary point of contact to DNA and that both dimers in tetrameric XL$(H3-H4)_2$ are shielded from DNA binding. As 18DNA can interact with XL$(H3-H4)_2$ (*Figure 4b*, lane 4, demonstrated by disappearance of the free DNA band into the well), this strongly suggests that two CAF-1 moieties remain bound to the XL$(H3-H4)_2$ tetramer, otherwise the exposed H3-H4 dimer in the constitutive tetramer XL$(H3-H4)_2$ would remain free to mediate DNA binding even with a complex with a mutated WHD. Because the tCAF-1•WTH3-H4•18DNA and the tCAF-1•DMH3-H4•18DNA complexes migrate similarly to the complex formed with XL$(H3-H4)_2$, which we know contains two tCAF-1 moieties (*Figure 2b–c*), and considerably slower than the tCAF-1•WTH3-H4 complexes without DNA (*Figure 4b*, lane 5–9, 488-H3-H4 red signal), we postulated that 18DNA promotes the association of two tCAF-1•WTH3-H4 moieties, and in this assembly histone tetramerization is not a requirement for complex formation.

To test this hypothesis, we performed in solution cross-linking studies. This allowed us to trap the assemblies without the potential artifact of the gel or column media, and also provided us with an opportunity to test our model with the FL CAF-1 complex, as the readout does not depend on DNA binding. Strikingly, titrating 18DNA into a FL CAF-1•WTH3-H4 complex strongly promoted the formation of a discrete high molecular weight assembly, which did not accumulate when the histones were omitted or when the Cac1 WHD was mutated (*Figure 4d* – *Figure 4—figure supplement 1a*). By analyzing these reactions through SEC-MALS, we confirmed that this species has a molar mass consistent with the assembly of (CAF-1)$_2$•(H3-H4)$_2$•18DNA (*Figure 4—figure supplement 1b*). In line with the cross-linking data, this species was not favored when the WHD was mutated, and the most abundant assembly was composed of a single CAF-1•H3-H4•18DNA complex (*Figure 4—figure supplement 1b*). This association was further supported by monitoring the formation of this band with

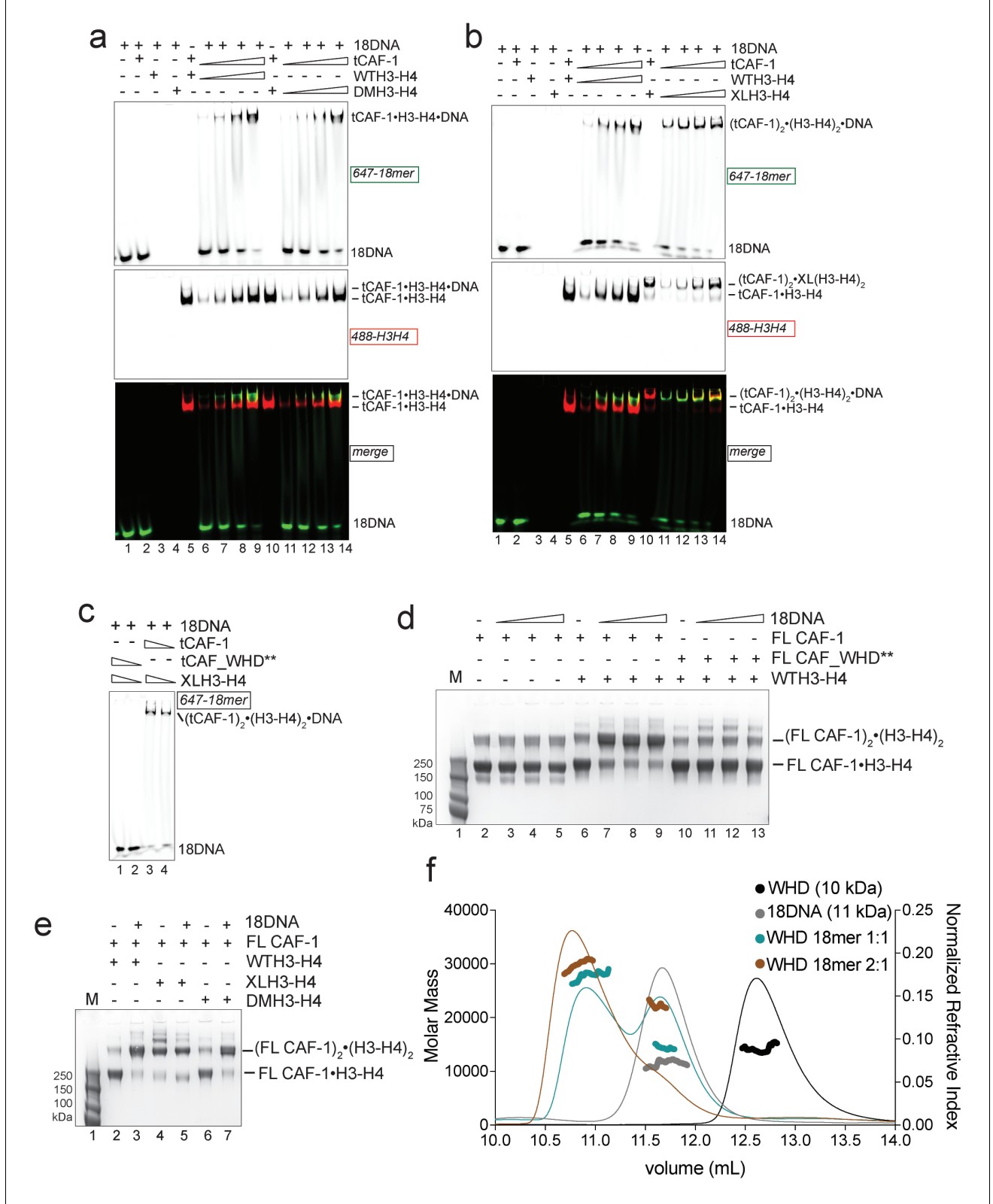

**Figure 4.** Two CAF-1•H3-H4 complexes associate on a short DNA fragment. (**a–b**) A ternary tCAF-1•histone•18DNA complex can be formed with DMH3-H4(**a**) or with XL(H3-H4)$_2$ (**b**). EMSA was performed with 100 nM 18 bp DNA (18DNA). tCAF-1 is titrated 0.37–0.75-1.5–3 µM, and is added at a 1:1 ratio to either DMH3-H4 (**a**) or XL(H3-H4)$_2$ (**b**). H3-H4 was calculated as a dimer, even with XL(H3-H4)$_2$ to the DNA. WTH3-H4 is shown as a control in both panels. (**c**) Both H3-H4 moieties in the XL(H3-H4)$_2$ are shielded from DNA binding in complex with tCAF-1. EMSA performed with 100 nM 18 bp

*Figure 4 continued on next page*

*Figure 4 continued*

DNA (18DNA). tCAF-1 or tCAF_WHD** in complex with XL(H3-H4)$_2$ [1 CAF-1 complex per 1 H3-H4 dimer] is at 1 and 3 µM. (d) In solution cross-linking experiments with 2 µM of either FL CAF-1 alone, FL CAF-1•H3-H4 complex, or FL CAF_WHD**•H3-H4 complex, and variable amounts of DNA (0-1-2-4 µM). DSS was added at 1 mM and incubated for 30 min, before quenching and analysis by SDS PAGE. Full gel image shown in *Figure 4—figure supplement 1a*. (e) In solution cross-linking experiments with DSS as in panel (d), but here FL CAF-1 was premixed with either WT, DMH3-H4 or XLH3-H4. DNA is at 4 µM. Full gel image shown in *Figure 4—figure supplement 1c*. (f) SEC-MALS experiment of the isolated WHD alone or in complex with 18DNA. In buffer containing 150 mM NaCl, the WHD is monomeric in absence of DNA, but on 18DNA it favors binding in a 2:1 stoichiometry (WHD to 18DNA). The protein/DNA elution traces (refractive index, RI) refer to the right y axis, the calculated molar masses refer to the left y axis.

The following figure supplement is available for figure 4:

**Figure supplement 1.** Control experiments for *Figure 4*.

XL(H3-H4)$_2$ and DMH3-H4, where a (CAF-1)$_2$•(H3-H4)$_2$ species is formed with XL(H3-H4)$_2$ even in absence of 18DNA (*Figure 4e – Figure 4—figure supplement 1c*), as predicted by our binding stoichiometry data (*Figure 2c* and *Figure 2—figure supplement 1f and h*). Importantly, titrating 18DNA into H3-H4 in absence of CAF-1 did not significantly stimulate the formation of (H3-H4)$_2$ tetramer, confirming that the association of two H3-H4 dimers under these conditions requires the presence of the CAF-1 chaperone (*Figure 4—figure supplement 1d*). Together, these data demonstrate that two FL CAF-1 complexes associate on a 18DNA via their WHD domains, but only do so when bound to histones.

To conclusively confirm that an 18DNA is sufficient to bridge two CAF-1 moieties, we measured the stoichiometry of the isolated Cac1 WHD when binding to 18DNA. In EMSA, we noticed that titration of an isolated WHD on longer DNA results in an alternate binding pattern that suggests a cooperative binding mode, while on 18DNA we only observed a single species (*Figure 4—figure supplement 1e*). SEC-MALS experiments demonstrate that this species contains two WHD domains, and that the binding of two WHDs to a 18DNA is preferred even at a 1:1 WHD to DNA stoichiometry (*Figure 4f*). Importantly, the Cac1 WHD elutes as a monomer on SEC-MALS in absence of DNA (*Liu et al., 2016*; *Zhang et al., 2016*) (*Figure 4f*). This indicates that two Cac1 WHDs cooperatively bind DNA, as observed for other WHDs (*Cornille et al., 1998*; *Gajiwala et al., 2000*; *Zheng et al., 1999*) and suggest that this can bring together two CAF-1•H3-H4 complexes on 18DNA.

Overall, our data support a model in which the interaction between DNA and the WHD is triggered by histone binding to CAF-1, and this promotes the juxtaposition of two CAF-1 complexes, each pre-loaded with an H3-H4 dimer. Thus, the DNA-mediated association of two histone-bound CAF-1 complexes may promote the formation of the (H3-H4)$_2$ tetramer on DNA.

## DNA length promotes (H3-H4)$_2$ deposition

How then is the (H3-H4)$_2$ tetramer deposited onto DNA? Our data indicate that the H3 α3 helix in the H3-H4 dimers bound to CAF-1 is available for tetramerization with another H3-H4 dimer (*Figure 2*). Moreover, we show that two CAF-1 complexes bind on either side of a (H3-H4)$_2$ tetramer, and that an obligate H3-H3' interaction is sufficient to stabilize their association (*Figure 2b*), suggesting no additional significant contacts between the two CAF-1 moieties. Because of its limiting length, 18DNA only accommodates WHD binding, but cannot fully wrap the histone tetramer, which allowed us to trap the ternary complex. Because this complex can also be formed with dimeric DMH3-H4 (*Figure 4a*, lanes 11–14), we concluded that histone tetramerization is not required for the formation of the 18DNA-containing intermediate.

We next used longer DNAs to see if we could monitor subsequent steps in tetrasome formation, to understand how the two H3-H4 dimers are joined and to demonstrate that the (tCAF-1)$_2$•(H3-H4)$_2$•18DNA complex is relevant in the nucleosome assembly reaction. We used a 33 bp DNA (33DNA) which is longer than required for WHD binding, but still shorter than the ~70 bp necessary for complete wrapping of the (H3-H4)$_2$. tCAF-1 forms a ternary intermediate with WTH3-H4 and DNA, shown by EMSA (*Figure 5a*, lanes 1–6). As observed with 18DNA, its formation is dependent on the integrity of the Cac1 WHD (*Figure 5a* lanes 7–10 and *Figure 5—figure supplement 1a*), supporting the idea that this domain mediates the primary interaction with DNA. Strikingly, the ternary complex containing 33DNA could not be stabilized with the constitutively dimeric DMH3-H4

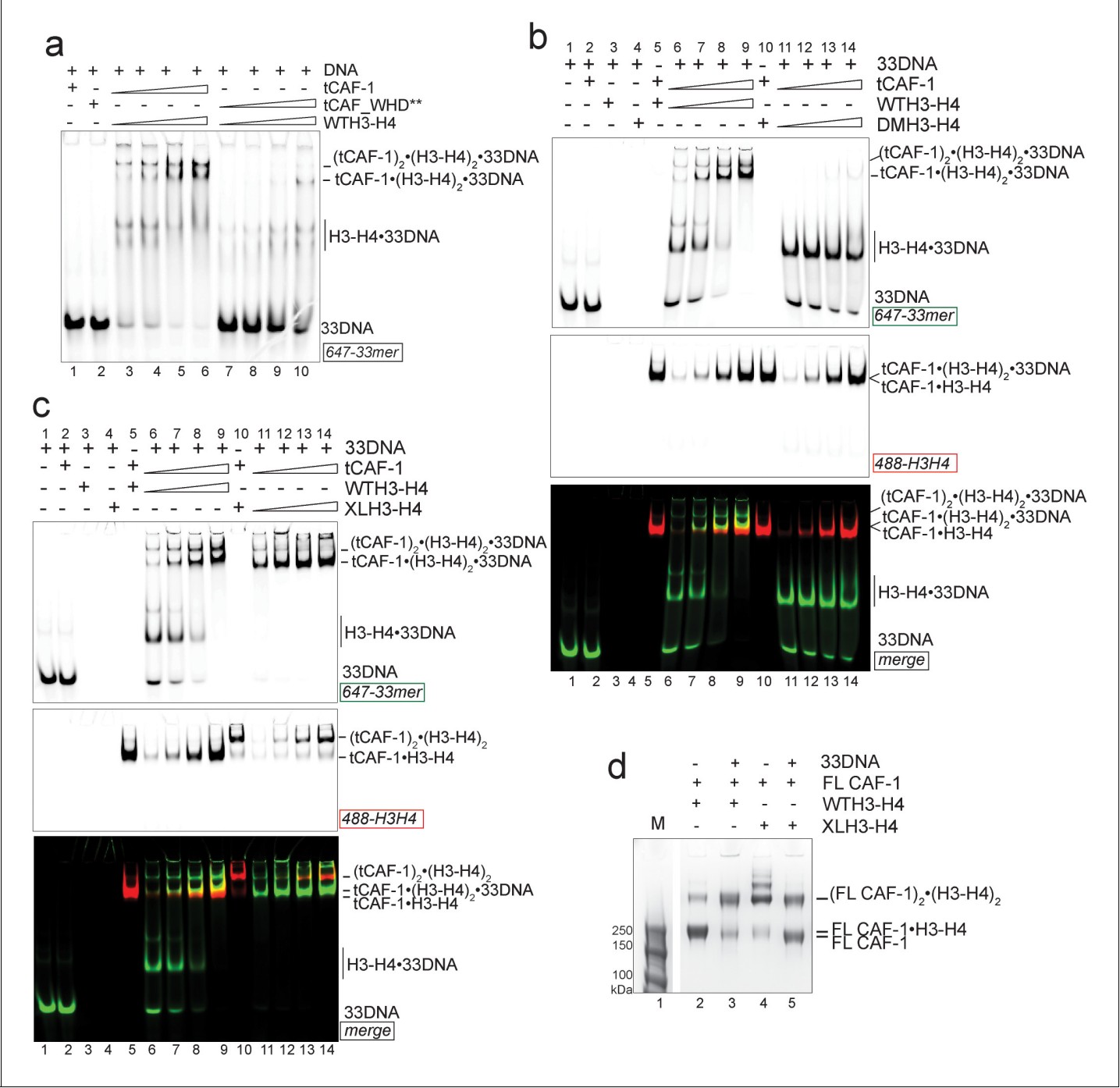

**Figure 5.** DNA of sufficient length sequesters (H3–H4)$_2$ from CAF-1. (**a**) EMSA performed with 100 nM 33mer DNA. tCAF-1 or tCAF-1_WHD** were titrated 0.37–0.75-1.5–3 µM, and were added with WTH3-H4 to the DNA. (**b**) EMSA performed with 100 nM 33 bp DNA (33DNA). tCAF-1 was titrated 0.37–0.75-1.5–3 µM, and was added with WT or DMH3-H4 (calculated as a H3-H4 dimer) to the DNA. (**c**) EMSA performed with 100 nM 33 bp DNA (33DNA). tCAF-1 was titrated 0.37–0.75-1.5–3 µM, and was added with WT or XL(H3-H4)$_2$ (calculated as a H3-H4 dimer) to the DNA. (**d**) In solution cross-linking experiments with 2 µM of FL CAF-1 premixed with either WT, DMH3-H4 or XL(H3-H4)$_2$, in presence of 4 µM of DNA. DSS was added at 1 mM and incubated for 30 min, before quenching and running of SDS PAGE. Full gel image is shown in *Figure 5—figure supplement 1c*.

The following figure supplement is available for figure 5:

**Figure supplement 1.** Control experiments for *Figure 5*.

(*Figure 5b*). This is surprising, as this histone mutant is sufficient to activate DNA binding by the WHD, as seen with the 18DNA (*Figure 4a*). These results demonstrate that histone tetramerization is required in this context. Hence, after the initial interaction via the Cac1 WHD, the formation of the ternary intermediate becomes dependent on the assembly of the $(H3-H4)_2$ tetramer with DNA of increasing length. This indicates that we are monitoring the step where the two H3-H4 dimers are joined to form the $(H3-H4)_2$ tetramer.

Consistent with the requirement for a histone tetramer, the ternary complex could be formed with $XL(H3-H4)_2$ (*Figure 5c*). With this histone isoform, we noticed that the 33DNA ternary complex migrated faster on the native gel than the previously identified $(tCAF-1)_2 \bullet XL(H3-H4)_2 \bullet 18DNA$ intermediate, suggesting a significant change in molar mass (*Figure 5c*). Consistent with this, when using 33DNA with $tCAF-1 \bullet XL(H3-H4)_2$, we saw a depletion of the peak corresponding to the $(tCAF-1)_2 \bullet XL(H3-H4)_2$ complex in SEC-MALS experiments (*Figure 5—figure supplement 1b*). Moreover, in cross-linking experiments with FL CAF-1 we noticed a similar trend, wherein the formation of a $(FL\ CAF-1)_2 \bullet XL(H3-H4)_2 \bullet DNA$ complex was disfavored with 33DNA (*Figure 5d*, lane 5 - *Figure 5—figure supplement 1c*). These observations support the idea that DNA of sufficient length partially destabilizes the CAF-1 •histone interaction.

Indeed, with a longer DNA (79 bp, 79DNA), suitable for complete tetramer wrapping, no intermediate could be formed and only the final assembly products, i.e. $(H3-H4)_2 \bullet DNA$ complexes, are detected (*Figure 5—figure supplement 1d*). In this context, the Cac1 WHD is still required for efficient histone discharge from the histone chaperone, even in presence of excess DNA (*Figure 5—figure supplement 1e*).

Overall, our data point to a model for CAF-1-mediated tetrasome formation in which H3-H4 dimer interaction with the CAF-1 complex releases and activates the Cac1 WHD to allow its interaction with DNA (*Figure 6a*). This interaction promotes the association of two histone-bound CAF-1 complexes, via cooperative DNA binding of the Cac1 WHDs (*Figure 6b*). Here, the two CAF-1 complexes join their H3-H4 dimers to form a $(H3-H4)_2$ tetramer that can then be sequestered by the

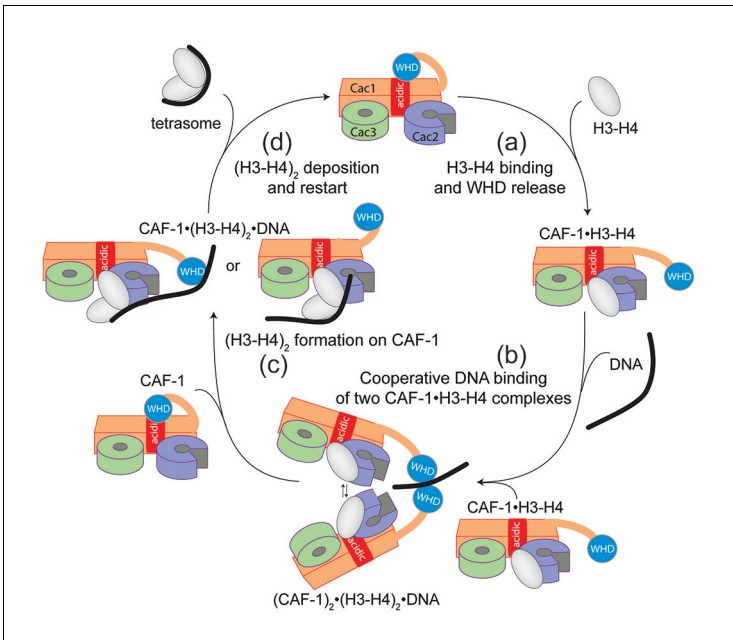

**Figure 6.** Model of the molecular mechanism of CAF-1 mediated tetrasome assembly. (**a**) The nucleosome assembly mechanism of CAF-1 is activated by H3-H4 binding, which releases the WHD domain from an intramolecular interaction with the acidic region on Cac1. (**b**) DNA binding promotes the association of two CAF-1•H3-H4 complexes to join the histones into a $(H3–H4)_2$ tetramer (**c**). In the presence of DNA of sufficient length, the $(H3–H4)_2$ histones are directly sequestered from CAF-1. (**d**) $(H3–H4)_2$ are transferred to the DNA to form the tetrasome, and the WHD re-binds to the now free acidic region, resulting in its dissociation from DNA.

DNA tethered by the WHD (*Figure 6c*). The exact mechanism by which DNA promotes histone unloading from CAF-1 remains yet to be determined. We propose two hypotheses. In one case, the DNA remains partially bound to one WHD as it begins to wrap around the tetrasome, in another scenario the DNA dissociates from the WHD as it becomes bound to the (H3-H4)$_2$ in proximity to the H3-H3′ region, where it begins to pry the histones off CAF-1 (*Figure 6c*). The restoration of the inhibitory intramolecular interaction between the Cac1 WHD and the histone binding region, which likely occurs as the histones are unloaded from CAF-1, may also contribute to their transfer to the DNA molecule (*Figure 6d*). Together, our data suggest that the Cac1 WHD not only contributes to the mechanism by merely binding DNA, but by actively bridging together two CAF-1 bound H3-H4 dimers. Furthermore, by directly competing with the histone binding region, the WHD may facilitate the transfer of (H3-H4)$_2$ from the chaperone to DNA.

## The WHD dictates nucleosome assembly

Our model predicts that the integrity of DNA binding by the WHD is a pre-requisite for tetrasome formation and hence nucleosome assembly. Indeed, in the NAQ assay, tCAF-1 complexes with a mutated or deleted WHD exhibit reduced nucleosome assembly activity (blue bars in *Figure 7a*). Notably, with these mutant tCAF-1 complexes we also observed a dose-dependent increase in protected DNA fragments around 100 bp rather than the 126–160 bp fragments observed with tCAF-1 and wild type CAF-1, indicative of the formation of subnucleosomal particles (*Figure 7b*). Similarly, in a native gel-based tetrasome assembly assay on either 147 or 79 bp DNA fragments, the WHD mutant complexes preferentially promoted formation of a band that migrates faster than the tetrasome (*Figure 7c* and *Figure 7—figure supplement 1a*, disome). This band corresponds to the deposition of a single H3-H4 dimer onto DNA, as seen using the constitutively dimeric DMH3-H4 (*Figure 7c*). This finding indicates that either abrogating WHD-dependent DNA binding, or preventing H3-H4 tetramerization, result in the same deposition product in vitro, a single H3-H4 dimer bound to DNA. This fully supports our model in which (H3-H4)$_2$ tetramerization or deposition cannot occur in absence of CAF-1 DNA binding (*Figure 6*) and indicates a direct role for the WHD in regulating the fidelity of the nucleosome assembly reaction.

Surprisingly, when testing the effect of WHD mutation and deletion in FL CAF-1, we observed no difference in activity in the NAQ assay (*Figure 7a*). A possible explanation for this observation is that in vitro in absence of PCNA, the DNA binding property of the N-terminal part of Cac1 may compensate for the mutation of the WHD. We therefore moved to an in vivo system to address if the WHD is required for FL CAF-1 activity in the cell. We first used a previously described assay that monitors epigenetic silencing of a telomere-proximal *URA3* reporter, a phenomenon which is dependent on the integrity of the Cac1 subunit (*Kaufman et al., 1997*). While WT Cac1 protein could completely rescue the growth phenotype on FOA plates, FL Cac1 WHD mutants did not, reflected by delayed growth (*Figure 7d* and *Figure 7—figure supplement 1b*). This indicates that the WHD domain is indeed important for Cac1 functions in the context of full-length Cac1 in vivo, in agreement with previous work (*Zhang et al., 2016*).

To specifically demonstrate that the phenotype observed with CAF-1 complex that is mutated in the WHD is due to an aberrant nucleosome assembly function of Cac1 in vivo, we took advantage of the fact that the length of Okazaki fragments generated during replication depends on proper nucleosome assembly by CAF-1 (*Smith and Whitehouse, 2012*; *Yadav and Whitehouse, 2016*). We analyzed Okazaki fragment length in yeast strains that carry mutations or deletions in the Cac1 WHD domain. Strikingly, Okazaki fragments purified from these strains do not exhibit the periodicity characteristic of a proficient replication-coupled nucleosome assembly pathway, but rather peak at length above 400 bp (*Figure 7e*). This phenotype is not as drastic as the one observed when we mutated the histone binding region (HA-Cac1_Nac, where the Cac1 acidic region is neutralized by a Gly-Ser-Leu peptide substitution; *Mattiroli et al., 2017*). Yeast strains harboring this mutant CAF-1 complex produced an Okazaki fragment pattern that resembled that of the Cac1 knock-out (*Figure 7e*). Unlike mutation of the Cac1 acidic region, Cac1 WHD deletion and mutation do not affect histone binding (*Figure 3—figure supplement 1c–e*) and hence these mutant complexes, unlike HA-Cac1_Nac, are still able to recruit H3-H4 at the replication fork. We speculate that passive histone diffusion from these CAF-1 constructs to the DNA may occur, resulting in residual and inefficient histone deposition. The clear difference in Okazaki fragment formation between the WT and the WHD mutations confirms a deficiency in timely and accurate replication-coupled nucleosome

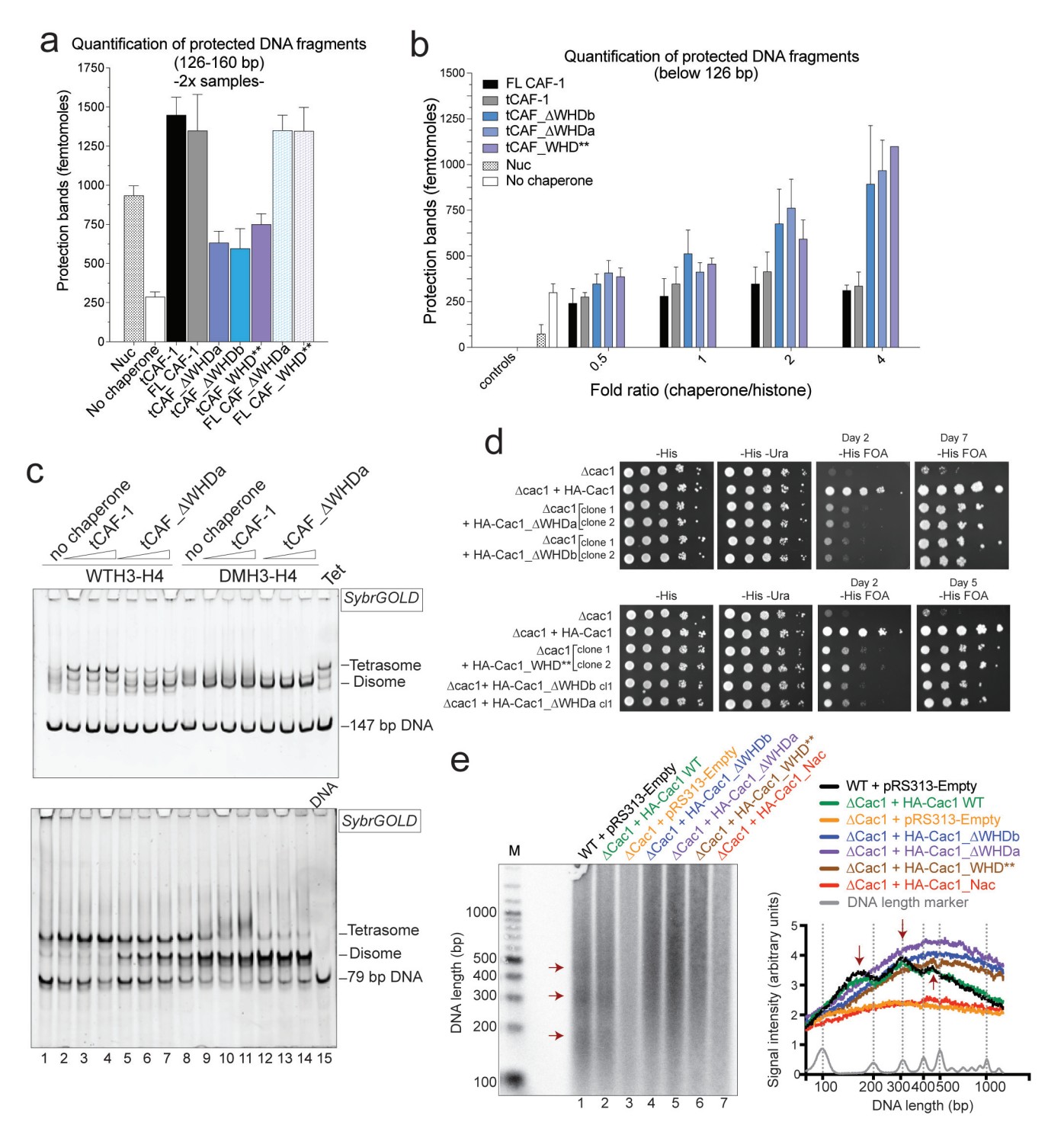

**Figure 7.** DNA binding by the Cac1 WHD is required for nucleosome assembly in vitro and in vivo. (a) Quantification of the nucleosome bands (126–160 bp) from the NAQ assays. Mean ± SD is shown from two or more independent measurements. For simplicity, we only show results from experiments with a 2-fold chaperone to histone ratio. The trends are identical for the 0.5-1-4 fold chaperone to histone ratio. The data use for this panel is included in *Figure 7—source data 1*. (b) Quantification of bands below 126 bp (sub-nucleosomal species) from the NAQ assays. The error bars indicate SD from at least three repeats. The data use for this panel is included in *Figure 7—source data 2*. (c) Tetrasome assembly assay on 147 bp (top panel) or 79 bp DNA (bottom panel) with tCAF-1 and the tCAF_ΔWHDa mutant, comparing WT and DMH3-H4 deposition. Assembly for the other WHD mutants is shown in *Figure 7—figure supplement 1a*. (d) Epigenetic silencing of a telomere-proximal URA3 reporter performed with yeast

*Figure 7 continued on next page*

*Figure 7 continued*

strains expressing the Cac1 mutation or deletion of the WHD. Samples were spotted at $0-10^1-10^2-10^3-10^4$ dilutions from a OD600 = 1 stock. (**e**) Okazaki fragment ends isolated from WT yeast strains, WHD mutant strains, and the acidic region mutant strain (Cac1_Nac, residues 397–431 were replaced with a Gly-Ser-Leu linker), as indicated, were radiolabeled and separated on a denaturing agarose gel. The right-hand panel depicts a normalized trace of signal intensity for each lane; DNA ladder is shown in light gray. Red arrows highlight the nucleosome-dependent length of the fragments in the positive control lanes (lane 1 and 2).

The following source data and figure supplement are available for figure 7:

**Source data 1.** Mutation of the Cac1 WHD inhibits nucleosome assembly in tCAF-1.
**Source data 2.** Mutation of the Cac1 WHD in tCAF-1 promotes the formation of sub-nucleosomal species.
**Figure supplement 1.** Control experiments to *Figure 7*.

assembly in vivo. These experiments show that the integrity of the Cac1 WHD is required for proper nucleosome formation on the lagging strand during DNA replication, fully supporting our model that places this domain at the heart of the CAF-1 nucleosome assembly mechanism (*Figure 6*).

## Discussion

Combined, our data suggest a complex mechanism for the nucleosome assembly function of CAF-1 (*Figure 6*). We propose a model that involves DNA binding of two histone-bound CAF-1 complexes, thereby joining two H3-H4 dimers which are then transferred to adjacent DNA, resulting in tetrasome assembly. Our data identifies the central role of the Cac1 WHD in orchestrating the assembly reaction through its histone-triggered DNA binding activity, and through promoting the association of two CAF-1 complexes, which culminates in $(H3-H4)_2$ tetramer and ultimately tetrasome formation.

Our work reveals that two CAF-1 complexes are needed for the formation of one tetrasome, with each complex contributing a single H3-H4 dimer. This provides direct evidence for previously proposed models based on in vivo observations (*Nakano et al., 2011*; *Tagami et al., 2004*). Nonetheless, previous studies have suggested that one CAF-1 promotes $(H3-H4)_2$ tetramer formation, based on FRET measurements (*Liu et al., 2012*, *2016*). Here, we have studied the CAF-1•H3-H4 stoichiometry using a variety of techniques, including label-free methods such as AUC, SEC-MALS and HX-MS. Our data indicate that CAF-1 has a single histone binding site that interacts with a H3-H4 dimer, although a second H3-H4 dimer (either with or without its own CAF-1 complex) can associate with the CAF-1-bound dimer. We propose that this mode of binding is essential in the nucleosome assembly mechanism to allow the juxtaposition of the histone dimers to form $(H3-H4)_2$ prior to deposition onto the DNA. This ability of CAF-1 to form different complexes with histones, depending on their concentration and conditions, may be responsible for the apparent differences in the literature (*Liu et al., 2012*, *2016*).

We demonstrate the central importance of the WHD in the histone deposition process. A recent study has identified cross-links between the Cac1 WHD and histones (*Liu et al., 2016*). While this was interpreted as a direct interaction between the DNA binding domain and the histones, our data suggest that this may be due to the physical proximity between the WHD and the histone binding region (*Figure 3b–d*). This interpretation is also corroborated by a different study, where the WHD was found to cross-link with Cac2 (*Kim et al., 2016*), another important component of the histone binding interface on CAF-1 (*Mattiroli et al., 2017*). Together, these findings suggest that the WHD remains in close proximity to H3-H4 on CAF-1. This could be an important structural aspect of the last step of the nucleosome assembly mechanism, where WHD-bound DNA sequesters the histones from the chaperone.

Our proposed model also demands that two CAF-1 complexes must be located in close proximity to be able to function together in tetrasome assembly. This implies that at least two CAF-1 complexes may simultaneously bind to one PCNA trimer. This is possible considering the presence of three potential binding sites for CAF-1 on a PCNA trimer, but will require further investigations. We cannot exclude that the interaction with PCNA may elicit additional structural rearrangement in

CAF-1 that could control the nucleosome assembly mechanism. The importance of the cross-talk between the CAF-1 nucleosome assembly function and PCNA is supported by the finding that Cac1 WHD and PCNA binding mutants exhibit a synergistic effect in vivo (*Zhang et al., 2016*).

Overall, our data suggest that a combination of relatively low affinity interactions (WHD•DNA, and H3•H3' in the H3-H4 tetramerization interface) work together to drive (H3-H4)$_2$ tetramer deposition onto DNA. This combinatorial effect ensures timely and accurate assembly reactions by bringing together chaperoned histones and DNA. Furthermore, on Cac1, the WHD is separated from the histone binding module by a flexible region (*Mattiroli et al., 2017*) which is post-translationally modified in a cell cycle dependent manner (*Holt et al., 2009*; *Jeffery et al., 2015*). These modifications may well play additional roles in fine-tuning the dynamic interactions of the WHD.

This study describes the mechanism of histone deposition by CAF-1, a reaction that has remained elusive for other histone chaperones. We demonstrate that an inhibitory intramolecular interaction poises CAF-1 for histone binding and deposition. H3-H4 dimer binding to CAF-1 activates the cascade of events required for its deposition, by unmasking the Cac1 WHD to bind DNA. This mechanism ensures that only histone-bound CAF-1 complexes are able to participate in the deposition process, thus optimizing the efficiency of tetrasome assembly and excluding inefficient participation of empty CAF-1 complexes. By mutating the WHD, we uncouple histone binding and deposition, and demonstrate that high affinity histone binding is not sufficient to sustain efficient nucleosome assembly. This suggests that not all proteins that bind histones (i.e. histone chaperones) are in fact efficient nucleosome assembly factors.

Finally, the observation that two CAF-1•H3-H4 complexes associate to form a tetrasome suggests that two independently chaperoned H3-H4 dimers are joined at the very last step of the deposition process, when the DNA is in proximity and available for sequestering the histones from CAF-1. While convincing evidence supports the conservative model, where parental (H3-H4)$_2$ are deposited in the same nucleosome following DNA replication (*Xu et al., 2010*), our proposed mechanism does not exclude the possibility of a semiconservative model for chromatin assembly coupled with DNA replication (*Tagami et al., 2004*). Further experiments are required to fully elucidate how CAF-1 complexes are paired in cells to understand the mechanisms underlying histone inheritance during cell division.

## Materials and methods

### Cloning and reagents

cDNA for the yeast CAF-1 complex were received from Paul Kaufman. These were cloned into the MultiBac vector for expression in insect cells. Cac1 was cloned into pACEBac1, Cac3 in pIDC, Cac2 in pIDS and these were recombined by Cre-Lox as described (*Bieniossek et al., 2008*). A His-tag was inserted at the C-terminal end of Cac2 for purification purposes. All complexes are prepared with a short C-terminal deletion of Cac2 (1–449), predicted to be disordered. Bacmids for expression in SF21 cells were prepared as previously described (*Bieniossek et al., 2008*). Mutations were introduced using Turbo Pfu polymerase (Roche) in a standard mutagenesis protocol. The Cac1 WHD cDNA was adjusted from the expression construct previously published (*Zhang et al., 2016*). The DNA used in the assembly and EMSA assays were based on the 601 Widom sequence and were prepared as previously described (*Dyer et al., 2004*).

### Protein preparations

*Xenopus laevis* histones were purified from *E.coli* cells as previously described (*Dyer et al., 2004*), and stored in 2 M NaCl at −80°C. Labeling of histone proteins was performed as previously described (*Muthurajan et al., 2016*). Specifically, we labeled H4 with Alexa-488 on T71C, and H2B with AttoN-647 on T112C prior refolding with the histone partner. Constitutive dimeric DMH3-H4 is prepared as WT histones and contains H3 C110E L126A I130A. Constitutive XL(H3-H4)$_2$ tetramers were prepared by incubating 25 μM histone (H3 C110A K115C – H4 WT) in 20 mM HEPES pH 7.5, 1 M NaCl, 1 mM EDTA with 50 μM BMOE (bismaleimidoethane) for 1 hr at room temperature. Cross-linking was quenched with a 5x stock of DTT (final concentration 10 mM DTT and 10 μM XL(H3-H4)$_2$). Cross-linking was assayed using a gel (samples were not boiled prior loading onto the gel).

CAF-1 was expressed in Sf21 cells and purified using a HisTrap column (GE) in buffer containing 50 mM TRIS 8.0, 600 mM NaCl, 5% glycerol, 10 mM imidazole, 5 mM BME (beta-mercaptoethanol), in presence of COMPLETE EDTA-free protease inhibitor (Roche), DNase I, 3 mM $CaCl_2$ and 3 mM $MgCl_2$. The complex was then loaded on a MonoQ column in buffer A (50 mM TRIS 8.0, 200 mM NaCl, 1 mM EDTA, 1 mM TCEP) and eluted with buffer B containing 1M NaCl. The protein was then injected into a size exclusion column (Superdex 200) in 30 mM Tris pH 7.5, 300 mM NaCl, 1 mM EDTA, 1 mM TCEP. For HX-MS studies, buffer containing 50 mM $KPO_4$, 150 mM NaCl, 5 mM DTT at pH 7.4 was used for the gel filtration step. Proteins were concentrated to 1–20 mg/ml and stored at −80°C in gel-filtration buffer. Mutants were purified as wild-type proteins; they behaved identically to wild type proteins during the purification procedure. CAF-1 complexes containing deletion or mutation of the acidic region in tCac1 were purified over a MonoS, instead of the MonoQ column with buffer A and B at pH 6.8. The WHD domain was expressed as a GST fusion and purified according to published protocols (*Zhang et al., 2016*).

## Hydrogen deuterium exchange coupled with mass spectrometry (HX-MS)

CAF-1 complexes and/or histones to be analyzed by HX-MS were prepared at 4 µM in HX buffer (50 $KPO_4$, 150 mM NaCl, 5 mM DTT pH 7.4). The CAF-1 complexes were gel filtered into the HX buffer, the histones were dialyzed overnight in the HX buffer to ensure that the final samples would not contain any additional buffer components that may result in buffer variability between proteins. HX reactions were set up by mixing 5 µl of the 4 µM stocks with 45 µl of deuterated HX buffer (prepared by dissolving in 99.9% $D_2O$ the lyophilized HX buffer) to result in a final 90% $D_2O$ concentration. Exchange was allowed to occur for 30 s, 1, 10, 30, or 60 min at 10°C. Exchange was quenched by adding 50 µl of ice cold quench buffer (25 mM succinic acid, 25 mM citric acid at pH 2), that brought the sample to pH 2.4. Pre-quenched control reactions were prepared by adding quench buffer prior to $D_2O$ buffer. The samples were immediately injected into a temperature controlled (0°C) Waters HDX Manager for online proteolysis at 12°C using an Poroszyme immobilized pepsin column (Life Technologies), followed immediately by a 3 min simultaneous peptide trapping and desalting step at 0°C using a Waters BEH UPLC C18 trap column, all with 100% solvent A (0.1% formic acid in water) flow at 100 µL/min. Peptides were then separated at 0°C using a Waters 100 mm BEH C18 analytical UPLC column and a linear 8% to 40% solvent B (0.1% formic acid in acetonitrile) gradient over 6 min, followed by a 1 min 40% B hold and subsequent ramp to 85% solvent B in 0.5 min using a Waters nanoAcquity UPLC and 40 µl/min flow rate. The UPLC was coupled directly with a Waters Synapt G2 HDMS q-TOF mass spectrometer operating in positive, MSe data acquisition mode. Samples were incubated and analyzed in a random order. Non deuterated, prequenched, 1 and 60 min samples were taken in triplicate.

PLGS 3.0 (Waters) was used to create an identified peptide list from non-deuterated datasets and DynamX 3.0 (Waters) performed the search for deuterated peptide ion assignments. All isotope assignments for each peptide in each charge state were manually verified. The weighted average mass of each peptide determined by DynamX was then used to calculate deuteron uptake which was converted to % of deuteration based on the number of maximum exchangeable amide protons (number of total amino acids – Pro). Data were corrected for artefactual in-exchange using the quenched experiment as previously reported (*Sours and Ahn, 2010*). No corrections for back-exchange were conducted due to the comparison of relative uptake amounts between bound and unbound states, which would remain unaffected by the back-exchange correction. Graph bars and uptake plots were prepared using Microsoft Excel and GraphPad prism. The HX-MS uptake values of all the peptides analyzed aresummarized in *Supplementary file 1*.

## Tetrasome assembly assays

The assays were carried out in buffer containing 25 mM TRIS pH 7.5, 150 mM NaCl, 1 mM EDTA, 0.02% Tween-20, 0.5 mM TCEP. CAF-1 was first diluted at different concentrations, normally a chaperone-histone ratio between 0.5 to 4 fold was used. Histones H3-H4 (100 nM tetramer concentration) were added, and the chaperone-histone mix was incubated at room temperature for 10 min. DNA was then added at 100 nM concentration. The reactions were incubated for 10–30 min (no differences were observed when incubating for longer time). Glycerol was added to a final

concentration of 10% v/v prior loading of the samples on a 6% PAGE gel, pre-run in 0.2x TBE buffer at 4°C. The gels were run for 70 min at 150 V at 4°C. Gels were stained with SybrGOLD for 10 min and imaged on a Typhoon FLA 9500 at 488 nm.

## NAQ assay (Nucleosome assembly and quantification)

The procedure for measuring nucleosome assembly activity in vitro with the nucleosome assembly and quantification (NAQ) assay is described in more detail at Bio-protocol (*Mattiroli et al., 2018b*). The assembly assay was carried out as described above containing 200 nM of 207 bp DNA, 200 nM (H3-H4)$_2$, 400 nM H2A-H2B and titration of CAF-1 (100-200-400-800 nM). After the assembly reaction, the samples were diluted to a DNA concentration of 50 nM in 100 µl digestion reactions. 25U of MNase enzyme was added in a final buffer containing 50 mM TRIS pH 7.9, 5 mM CaCl$_2$. After incubation at 37°C for 10 min, the reactions were quenched with 10 µl of 500 mM EDTA, pH 8. The DNA was then purified using a modified protocol of the MinElute kit from QIAGEN. 550 µl of PB buffer and 10 µl of 3 M sodium acetate were added to each sample and they were incubated at room temperature for 10 min. At this point, 50 ng of DNA loading control (or reference band, a 621 bp DNA fragment) was added to each tube. The samples were applied to the MinElute spin column and washed as prescribed by QIAGEN. The DNA was eluted with 10 µl of water. 1 µL of the eluate was used to load a DNA 1000 chip on the Bioanalyzer machine (Agilent), and 2.5 µl were loaded on a 10% PAGE gel. The gel was run for 45 min at 200 V in 0.5x TBE buffer at room temperature. Gels were stained with SybrGOLD for DNA and imaged on a Typhoon FLA 9500 (GE). The Bioanalyzer data were analyzed using the Agilent Expert 2100 software. The reference band was corrected for the proper size (621 bp) and the calculated molarity values were used to normalize all other bands present in the lane. The normalized values were used in the quantification and comparison. The signal threshold was set at 20 RFU. Nucleosome signal was calculated from bands ranging between 126–160 bp in length, based on the digestion of salt-reconstituted nucleosomes. Bands above or below this range were quantified separately. At least three independent repeats were performed per experimental condition.

## Fluorescence polarization experiments

Fluorescence Polarization assays were carried out in 25 mM TRIS pH 7.5, 300 mM NaCl, 5% glycerol, 1 mM EDTA, 0.01% NP-40, 0.01% CHAPS, 1 mM DTT (added fresh). Binding reactions were prepared by mixing 5 nM of Alexa488-labeled H3-H4 dimer and increasing amounts of CAF-1. Binding data were measured using a BioTek Synergy H2 plate reader. The data was analyzed and plotted using Microsoft Excel and GraphPad Prism. The competition assay shown in *Figure 1—figure supplement 1e* was done using fluorescence quenching rather than polarization. Representative curves are shown from one experiment (three independent measurement). The same conclusions were drawn from at least two other replicates, each containing three independent measurement per data point.

## FRET-based stoichiometry experiments (Job plot)

FRET-based stoichiometry measurements of protein complexes in vitro are described in more detail at Bio-protocol (*Mattiroli et al., 2018a*). FL or tCAF-1 were freshly labeled with an equimolar amount of AttoN-647 dye for 1 hr at 4°C. The reactions were quenched with 10 mM DTT. The protein was purified from the free dye on a PD-10 desalting column in buffer 20 mM TRIS pH 7.5, 300 mM NaCl, 1 mM EDTA, 1 mM TCEP. CAF-1 complexes were concentrated and mixed with Alexa488-labeled H3-H4 at constant total protein concentration of 150 nM, by inverse titrations of either species. Titrations containing only one labeled protein were used to correct the FRET signal. These were measured on a Typhoon or on a BMG ClarioStar plate reader. Dimer concentration was used for WT and DMH3-H4, while tetramer concentration was used for XL(H3-H4)$_2$. Representative curves are shown from one experiment (two independent measurement). The same conclusions were drawn from at least one other replicate, containing two independent measurement per data point.

## Sedimentation velocity analytical UltraCentrifugation (SV-AUC)

SV-AUC experiments were carried out in a final buffer containing 20 mM TRIS pH 7.5, 300 mM NaCl, 1.6% glycerol, 1 mM EDTA, 1 mM TCEP. Samples were prepared at 4°C at a concentration of 4 µM

CAF-1 (4 or 8 µM H3-H4 dimers) and run in a Beckman XL-A or XL-I centrifuges at 37,000 rpm at 20°C. Absorbance at 280 nm was monitored. The data were analyzed using UltraScan III.

## Size exclusion chromatography in line with multi-angle light scattering (SEC-MALS)

A Superdex 200 10/300 GL (GE) was mounted in line with a Optilab DSP and a DAWN Eos detectors (Wyatt). The runs were performed in 20 mM TRIS pH 7.5, 150 mM NaCl, 1 mM EDTA, 1 mM TCEP at room temperature. 100 µL of protein sample at 15 or 100 µM (CAF-1 or WHD samples respectively) were injected at 0.4 ml/min, after being spun down at highest speed for 10 min. Data analysis was done using the ASTRA software (Wyatt) and GraphPad Prism was used to prepare figures. The WHD-18DNA experiments were run on a Superdex 75 10/300 column in line with a DAWN HELEOS II light scattering and a Optilab rEX refractive index detectors (Wyatt).

## EMSA experiments

Ternary complex formation was set up in 25 mM TRIS pH 7.5, 150 mM NaCl, 1 mM EDTA, 0.02% Tween-20, 0.5 mM TCEP. Proteins were incubated 10 min (3 µM of a 1:1 ratio of tCAF-1 and H3-H4 dimer concentration, or titration as described in figure legend) before addition of DNA (100 nM, titration of DNA to higher concentration didn't result in higher amount of ternary complex). Histone concentrations refer to histone dimers, even in the case of $XL(H3-H4)_2$. 10% final concentration of glycerol was added before loading the samples into a 6% PAGE, pre-run in 0.2x TBE buffer at 150V. Gels were either scanned for fluorescence directly (AttoN647-labeled 18 or 33 bp and labeled histones) or first stained with SybrGOLD and then scanned on a Typhoon FLA 9500.

## In solution cross-linking

2 µM of a 1:1 ratio of CAF-1 and H3-H4 dimer were incubated for 30 min in 20 mM HEPES pH 7.5, 75 mM NaCl, 1 mM DTT. Histone concentrations refer to histone dimers, even in the case of $XL(H3-H4)_2$. Then, DNA or buffer control was titrated at 1-2-4 µM (or just 4 µM for the single concentration reaction). After 10 min, 1 mM DSS (disuccinimidyl suberate) was added and the reaction was left for 30 min at room temperature. The reactions were quenched with a final concentration of 70 mM TRIS pH 7.5. SDS loading buffer was immediately added and the samples were run on a NUPAGE 4–12% gel in MES buffer.

## Yeast heterochromatin maintenance in vivo assay

The endogenous *CAC1* locus was cloned into a pRS313 vector (the fragment contained 659 bp upstream and 731 bp downstream the Cac1 ORF). An HA-tag was introduced in the 5' end of the ORF. This construct was used to transform the PKY106 strain obtained from PD Kaufman (*Kaufman et al., 1997*). Mutants were generated using site-directed mutagenesis and were treated as the WT sample. The empty pRS313 vector was used as a control. Transformed PKY106 strains were grown in synthetic media lacking histidine (-His). Clones were amplified overnight and then diluted in the morning and grown to an OD600 = 0.7–0.8. After washing the cells with water and then resuspending them to an OD600 = 1, four 10-fold dilutions from the initial stock were prepared. The undiluted sample and these dilutions were spotted on plates containing -His or -His-Ura media as controls, and -His supplemented with 1 mg/ml 5-Fluoroorotic Acid (FOA) for growth selection. Plates were left at 30°C for 2 days and then left at room temperature for up to the seventh day. To validate HA-Cac1 expression, 7 ml of cultures were harvested at OD600 = 0.7–0.8 and washed in water. The pellet was boiled for 3 min. 50 µl of PBS buffer containing 1 mM TCEP and COMPLETE EDTA-free protease inhibitors was added to the pellet. Cells were lysed using glass beads and the lysate was then spun down. The supernatant was loaded on a SDS PAGE gel and transfer to a PVDF membrane. The blot was probed with anti-HA antibodies (Abcam ab9110 RRID:AB_307019).

## Okazaki fragment assay

Budding yeast strains yIW347 (wild-type, WT) and yIW397 (Δcac1) (*Yadav and Whitehouse, 2016*) carrying degron-tagged doxycycline-repressible alleles of CDC9 were transformed with empty vector pRS313. yIW397 was also transformed with pRS313,HA-Cac1 construct described above. All transformants were verified and maintained by growth on YC-HIS minimal media. For the Okazaki

fragment preparation, all strains were grown at 30°C in YC -HIS liquid medium supplemented with 2% glucose. 50 ml cultures were used for labeling experiments. At OD600 ~0.3, cells were harvested and re-suspended into 50 ml YEP medium and doxycycline was added to final concentrations of 40 mg/l and the culture shaken at 30°C for 2.5 hr. Genomic DNA was prepared from spheroplasts, radio-labeled, and visualized as described previously (*Smith et al., 2015*).

## Acknowledgements

We thank P Kaufman for providing the cDNAs of the CAF-1 subunits and the PKY106 strain used in the in vivo silencing assays, I Berger for the MultiBac vectors, Y Liu for the WHD expression construct, the Biofrontiers sequencing facility at CU Boulder for Bioanalyzer runs, and S D'Arcy for the optimization of the XL(H3-H4)$_2$ crosslinking procedure. Histones were obtained from the Protein Expression and Purification Facility at Colorado State University. We thank J Rudolph and K Zhou for critical reading of the manuscript. FM is funded by EMBO (ALTF 1267–2013) and the Dutch Cancer Society (KWF 2014–6649). Research in the Luger lab is funded by the Howard Hughes Medical Institute and NIH (GM067777). NIH funded the work of NGA (R01 GM114594), IW (R01 GM102253) and the CU Boulder Central Analytical Laboratory and Mass Spectrometry facility (S10 RR026641). CAR and LAS are funded by the National Science Foundation (MCB-1330019).

## Additional information

### Funding

| Funder | Grant reference number | Author |
|---|---|---|
| Howard Hughes Medical Institute | Investigator | Karolin Luger |
| National Institute of General Medical Sciences | GM067777 | Karolin Luger |
| European Molecular Biology Organization | ALTF 1267-2013 | Francesca Mattiroli |
| KWF Kankerbestrijding | 2014-6649 | Francesca Mattiroli |
| National Science Foundation | MCB-1330019 | Laurie A Stargell |
| National Institute of General Medical Sciences | GM114594 | Natalie G Ahn |
| National Institute of General Medical Sciences | GM102253 | Iestyn Whitehouse |

The funders had no role in study design, data collection and interpretation, or the decision to submit the work for publication.

### Author contributions

FM, Conceptualization, Data curation, Formal analysis, Supervision, Funding acquisition, Validation, Investigation, Visualization, Methodology, Writing—original draft, Project administration, Writing—review and editing; YG, Conceptualization, Data curation, Formal analysis, Validation, Investigation, Visualization, Methodology, Writing—original draft, Writing—review and editing; TY, Data curation, Formal analysis, Visualization, Methodology; JLB, Data curation, Formal analysis, Supervision, Methodology; MRH, Data curation, Formal analysis, Validation; ESF, YL, Data curation, Formal analysis; CAR, Data curation, Formal analysis, Methodology; LAS, Formal analysis, Supervision, Methodology; NGA, Formal analysis, Supervision, Funding acquisition, Methodology; IW, Data curation, Formal analysis, Supervision, Funding acquisition, Validation, Methodology; KL, Conceptualization, Resources, Data curation, Formal analysis, Supervision, Funding acquisition, Validation, Investigation, Visualization, Writing—original draft, Project administration, Writing—review and editing

### Author ORCIDs

Francesca Mattiroli, http://orcid.org/0000-0002-1574-7217
Yajie Gu, http://orcid.org/0000-0002-1514-1477

Iestyn Whitehouse, http://orcid.org/0000-0003-0385-3116

Karolin Luger, http://orcid.org/0000-0001-5136-5331

## Additional files

### Supplementary files

• Supplementary file 1. HX-MS data and uptake plots for tCAF-1 alone and in complex with WT or DMH3-H4.

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
