## [Decision Letter]

Thank you for submitting your article "DNA-mediated association of two histone-bound CAF-1 complexes drives tetrasome assembly in the wake of DNA replication" for consideration by *eLife*. Your article has been reviewed by three peer reviewers, and the evaluation has been overseen by a Reviewing Editor and John Kuriyan as the Senior Editor. The reviewers have opted to remain anonymous.

The reviewers have discussed the reviews with one another and the Reviewing Editor has drafted this decision to help you prepare a revised submission.

Summary:

The reviewers find that the paper describes an interesting study of the mechanism of histone (H3/H4)_2_ tetrasome deposition by Chromatin Assembly Factor-1, illuminating dependency relationships among protein domains involved in binding histones and DNA.

Central conclusions:

1) In the absence of the N-terminal region of the large Cac1 subunit, yeast CAF-1 displays little affinity for DNA, but instead binds histones H3/H4.

2) Histone binding stimulates DNA binding by the winged-helix domain (WHD) of the large CAF-1 subunit.

3) Binding of the WHD of CAF-1 to short DNA fragments facilitates H3/H4 tetramerization (while still bound to two CAF-1 proteins).

4) Different lengths of DNA lead to different requirements for complex formation, with longer DNAs requiring tetramerization of H3/H4 and favoring the release of CAF-1.

5) Deletion of the WHD domain causes defects in Okazaki fragment length and heterochromatic gene silencing, indirect measures of nucleosome deposition. In vitro, WHD mutations cause modest defects in nucleosome deposition, but only if the N-terminal DNA binding domain of the large CAF-1 subunit is missing.

Essential revisions:

A) The following points describe concerns or questions raised by the reviewers, where additional experiments would be helpful. Please consider whether these experiments can be done. If not, please explain why and clarify the issues in the revised manuscript.

1) The authors should address whether the deletion of the WHD alters the kinetics or fidelity of the nucleosomes assembly reaction. Currently, it is unclear whether this DNA binding is only involved in localization of CAF-1 to the substrate DNA or whether binding to the DNA also involves a facilitated transfer of the DNA from CAF-1 to the newly formed nucleosome/tetrasome. Would binding to PCNA substitute for the role of the WHD?

2) The complex with tCAF-1, H3/H4, and DNA reported in Figure 4 is particularly intriguing. Some structural or dynamic data, perhaps from HX, using this complex would add substantial insight into the authors' model for the mechanism of tetrasome assembly by CAF-1.

3) Figure 4, Lane 3. XL-H3-H4 seems to have the same effect as DNA in promoting (FL CAF-1)_2_.(H3-H4)_2_ formation. This suggests that the formation of this species requires cross-linking between H3-H4 dimers by DSS that may be promoted by the 18-mer of DNA. DSS cross-linking assays with H3-H4 +/- 18mer-DNA would be a good control for this.

4) Figure 7. The WHD mutants seem to work very differently than the empty vector control. Why is this? The authors should compare to other mutants (e.g. in the acidic region required for histone binding and/or in the PCNA interaction domain). Also, Figure 7 – a different color should be used to illustrate the WT sample and the DNA ladder.

B) The following points should be clarified in the revised manuscript, and additional experimental work is not essential.

1) The text should be amended according to the concerns in sections a-d below.

a) The main claims of the paper are stated with a strong tendency for hyperbole and these claims should be toned down. For example:

The authors claim their study is the "…first direct mechanism for histone deposition by a histone chaperone…". "Direct" seems too 'user-defined' in this instance. And "first" will certainly be contested by others in the field.

In the Abstract: "here we elucidate the CAF1:H3-H4 binding mode and mechanism of nucleosome assembly." This work provides insight to the process, but the matter is far from solved, and Figure 7 indicates that there may be multiple H3/4 binding modes and assembly mechanisms.

Discussion: "This suggests that not all proteins that bind histones (i.e. histone chaperones) are in fact nucleosome assembly factors." This is already well appreciated in the field and not addressed in the current work.

b) There are a number of statements that should be better referenced. For example:

In several places the authors refer to H3-H4 binding sites in CAF-1 (Cac1 acidic domain and Cac2) and refer to their own related but unpublished manuscript. It would be more appropriate to cite Liu et al., 2016 and Kim et al., 2016 unless their data do not agree with those reported in the related manuscript by the authors.

Results: "it appears that the primary role of CAF-1 is to promote the formation of an ordered (H3-H4)_2_-DNA complex, the tetrasome." This statement corroborates previous work that should be referenced.

c) The authors need to do a better job of explaining/synthesizing earlier results/conclusions from others (e.g. *eLife* paper from Liu et al. 2016). Likewise, the conclusions of this paper seem to contradict those of Winkler et al. 2012 from the same laboratory. Yet this paper is not cited in the places where the authors refer to previous studies showing CAF-1 promoting H3-H4 tetramer formation (Liu et al. 2012, 2016) (e.g., Introduction, fifth paragraph). The authors should clarify how the new work differs from previous results both their own and other labs, and discuss potential reasons (different assays, buffers, etc.) why different conclusions have been reached previously. This discussion may become quite technical, but it is essential to clarify the advances made here and distinguish them from previous studies. The current discussion misses the opportunity to clarify how the authors think previous experiments 'got it wrong' on issues of stoichiometry and/or to synthesize the findings coming from other studies.

d) The highly speculative discussion of the authors' ideas on how stoichiometry might yield inheritance of histones at replication should be shortened/eliminated as it is not clear that the current studies provide new insights into this process. Indeed, it is not clear that CAF-1 functions in this process at all (CAF-1 may only function during new histone deposition).

2) The authors use HX-MS experiments to probe changes in CAF-1 conformation upon H3-H4 binding. Similar experiments were carried out by Liu et al. 2016. The authors should compare the data sets and highlight potential differences or, if there is not substantial new information, remove the figure.

3) The model presented in Figure 6 predicts that the WHD interacts with the acidic region. Can the authors confirm this with direct binding assays or is there already evidence for this in the recent crosslinking studies of CAF-1 (Liu et al., 2016; Kim et al., 2016)?

4) As mentioned by the authors, crosslinking experiments showed contacts between the WHD and histones. This seems at odds with the model predicted in this paper. Could it be that the WHD aids histone binding for non-chromatin bound CAF-1, and in vivo only switches to bind DNA once the N-terminal region has recruited CAF-1 to PCNA and new DNA? The authors should consider this possibility.

---

## [Author Response]

Essential revisions:

A) The following points describe concerns or questions raised by the reviewers, where additional experiments would be helpful. Please consider whether these experiments can be done. If not, please explain why and clarify the issues in the revised manuscript.

1) The authors should address whether the deletion of the WHD alters the kinetics or fidelity of the nucleosomes assembly reaction.

The current experimental setup does not allow us to follow the kinetics of the reactions, due to the use of gel-based assays. We are therefore unable to comment on the effect of the WHD on kinetics. However, our data show that in absence of the WHD, the fidelity of the reaction is drastically compromised (Figure 7 and Figure 7—figure supplement 1). With a mutated or deleted WHD, tetrasome assembly is disfavored and formation of a disome is preferred (Figure 7 and Figure 7—figure supplement 1). We propose that fidelity of tetrasome assembly is primarily due to the cooperative mode of DNA binding of the WHD domain (Figure 4 and Figure 4—figure supplement 1). We added a comment about this in the first paragraph of the subsection “The WHD dictates nucleosome assembly”.

Currently, it is unclear whether this DNA binding is only involved in localization of CAF-1 to the substrate DNA or whether binding to the DNA also involves a facilitated transfer of the DNA from CAF-1 to the newly formed nucleosome/tetrasome.

We first want to emphasize that because the WHD binds DNA in a cooperative manner (as shown in Figure 4), its function goes beyond simply tethering the complex to the DNA. The assembly mechanism is based on our finding that during DNA binding, two CAF-1/H3-H4 complexes are joined to promote product formation (Figure 4 and Figure 4—figure supplement 1, and Figure 6).

However, to further drive home this point, we have now performed competition experiments where we titrate excess DNA to solutions containing a constant concentration of tCAF-1•H3-H4 complexes. In these experiments, DNA titration stimulates histone discharge from CAF-1 in a manner that is WHD-dependent, supporting the notion that the effect of the cooperative DNA binding of the WHD persists even in presence of excess DNA (new Figure 5—figure supplement 1, subsection “DNA length promotes (H3-H4)_2_ deposition”, fourth paragraph).

Furthermore, we propose that the return of the WHD to its auto-inhibited position on the Cac1 acidic region may contribute to a facilitated transfer of the histones from CAF-1 to DNA. Although this would be very challenging to prove experimentally, we think it is worth pointing out in light of this reviewer’s comment. We have added these suggestions in the last paragraph of the aforementioned subsection.

Would binding to PCNA substitute for the role of the WHD?

We indeed tested whether PCNA could stimulate the association of two CAF-1/H3-H4 complexes, using in solution cross-linking experiments. Here, we see that PCNA interacts with both FL CAF-1 and FL CAF_WHD**, as expected because the WHD mutations do not affect the PCNA-binding region located in the N-terminal part of Cac1. Our experiments clearly show that PCNA, in absence of DNA, does not stimulate the association of two CAF-1 complexes, and hence PCNA itself does not seem to substitute for the role of the WHD-dependent cooperative binding to the DNA (Figure 8). This is in line with in vivo data from the Zhang lab (Zhang et al., 2016 NAR) showing that PCNA interaction and WHD mutation have a synergistic effect, and therefore contribute to CAF-1 function through different mechanisms.

Author response image 1.In solution cross-linking studies of the complexes formed by FL CAF-1/H3-H4 in presence of 1 18 bp DNA (18DNA) or PCNA.CAF-1/H3-H4 was kept at 2 µM, 18DNA and PCNA were added at 4 and 8 µM, respectively. The whole gel is shown on the left, the close-up is shown on the right.**DOI:**
http://dx.doi.org/10.7554/eLife.22799.020

However, we note here and in the Discussion that a comprehensive analysis of the role of PCNA in this mechanism is required to make a final assessment of its contribution. For example, DNA loading of PCNA may be a requirement to fully recapitulate the system; but in our opinion this exceeds the scope of this manuscript. For these reasons, we have not included Figure 8 in the final version of the manuscript, but only include it for the reviewer’s information.

Overall, our data support a complex role for the WHD that extends beyond DNA binding, and that involves the cooperative association of two complexes, and we propose that this may actively facilitate histone transfer. These activities may not be substituted by the interaction with PCNA.

2) The complex with tCAF-1, H3/H4, and DNA reported in Figure 4 is particularly intriguing. Some structural or dynamic data, perhaps from HX, using this complex would add substantial insight into the authors' model for the mechanism of tetrasome assembly by CAF-1.

We agree with the reviewer that this complex is interesting and that any structural information would be extremely valuable. We have not yet been able to perform such in-depth analysis, including using HX-MS, because we have had difficulties isolating a stoichiometric, stable and homogenous sample of this ternary complex. Attempting HX-MS or other analyses with complexes that are not stoichiometric or stable will impede valid interpretation of the results. More precisely, because HX-MS results report a measurement of the entire population, the simultaneous analysis of a heterogeneous mixture of species that represents variable conformations for incompletely bound vs. unbound states can lead to results that cannot be distinguished. We are currently developing strategies to increase our molecular understanding of this complex, but this will require a considerable investment of effort and time that we think is beyond the scope of this manuscript.

However, to further strengthen the characterization of the reaction intermediate, we have now included new SEC-MALS experiments to confirm that the cross-linked complex presented in Figure 4 (lane 9) has indeed a molecular mass consistent with the stoichiometry of (CAF-1)_2_•(H3-H4)_2_•18DNA (new Figure 4—figure supplement 1). Formation of this complex is favored in presence of wild-type FL CAF-1, while the WHD mutated complex (FL CAF_WHD**) favors the single CAF-1•H3-H4 complex, in line with our other data (i.e. Figure 3 and Figure 4, lane 13).

3) Figure 4, Lane 3. XL-H3-H4 seems to have the same effect as DNA in promoting (FL CAF-1)_2_.(H3-H4)_2_ formation. This suggests that the formation of this species requires cross-linking between H3-H4 dimers by DSS that may be promoted by the 18-mer of DNA. DSS cross-linking assays with H3-H4 +/- 18mer-DNA would be a good control for this.

It is indeed true that crosslinking of the histones (by means of using XL(H3-H4)_2_ prior to addition of DNA) promotes the association of two CAF-1 complexes, as shown in Figure 4, lane 4. We have now performed the suggested control experiment (shown in new Figure 4—figure supplement 1), which clearly shows that addition of 18 bp DNA to H3-H4, in absence of CAF-1, does not promote histone tetramerization. This result confirms our hypothesis that the DNA-mediated association of two CAF-1 complexes, each with its histone cargo, precedes the joining of the CAF-1-bound H3-H4 dimers.

4) Figure 7. The WHD mutants seem to work very differently than the empty vector control. Why is this? The authors should compare to other mutants (e.g. in the acidic region required for histone binding and/or in the PCNA interaction domain). Also, Figure 7 – a different color should be used to illustrate the WT sample and the DNA ladder.

We have now added the suggested control (Cac1 with a neutralized acidic region, Cac1_Nac, shown in the updated Figure 7) and we have changed the color code.

The reviewer is correct in that there is a difference in Okazaki fragment distribution between the KO and the WHD mutants. Interestingly, the suggested control mutant Cac1_Nac behaves like a Cac1 deletion strain (updated Figure 7). In this mutant, the acidic region has been replaced by a neutral linker, leading to defective histone binding (binding affinity for H3-H4 in vitro is 10-fold lower, shown in the related manuscript).

We believe that this difference is because CAF-1 WHD mutants bind H3-H4 with the same affinity and stoichiometry as WT CAF-1 and hence they recruit H3-H4 to the replication fork through the interaction between CAF-1 and PCNA which is not affected by the WHD mutations (shown in Figure 8). This may result in passive transfer of H3-H4 onto the nascent DNA, which is likely less efficient than nucleosome assembly mediated by cooperative interaction of two CAF1 moieties through their WHD. In contrast, the Cac1_Nac mutant, which exhibits greatly reduced binding affinity for H3-H4, may fail to efficiently recruit H3-H4 to the replication fork, resulting in the more drastic phenotype that resembles that of the Cac1 KO strain.

Importantly, the lack of periodicity of the Okazaki fragments purified from WHD mutant strains, together with their extended length (above 400 bp) strongly suggest that timely and controlled replication-coupled nucleosome assembly is compromised in vivo, in line with our model in which the Cac1 WHD is central in orchestrating the accurate histone deposition mechanism.

We have clarified these issues in the last paragraph of the subsection “The WHD dictates nucleosome assembly”.

B) The following points should be clarified in the revised manuscript, and additional experimental work is not essential.

1) The text should be amended according to the concerns in sections a-d below.

a) The main claims of the paper are stated with a strong tendency for hyperbole and these claims should be toned down. For example:

The authors claim their study is the "…first direct mechanism for histone deposition by a histone chaperone…". "Direct" seems too 'user-defined' in this instance. And "first" will certainly be contested by others in the field.

We have now rephrased these statements in the Abstract and in the text (Introduction, last paragraph and throughout the Discussion).

In the Abstract: "here we elucidate the CAF1:H3-H4 binding mode and mechanism of nucleosome assembly." This work provides insight to the process, but the matter is far from solved, and Figure 7 indicates that there may be multiple H3/4 binding modes and assembly mechanisms.

We have now toned down this sentence in the Abstract.

Discussion: "This suggests that not all proteins that bind histones (i.e. histone chaperones) are in fact nucleosome assembly factors." This is already well appreciated in the field and not addressed in the current work.

While this reviewer clearly appreciates this distinction, to our knowledge this has not been explicitly made in literature. Therefore, we prefer to leave this comment in the Discussion. Although this distinction is not directly addressed by our work, we think it makes an important point that will further refine our understanding of histone chaperones.

*b) There are a number of statements that should be better referenced. For example:*

In several places the authors refer to H3-H4 binding sites in CAF-1 (Cac1 acidic domain and Cac2) and refer to their own related but unpublished manuscript. It would be more appropriate to cite Liu et al., 2016 and Kim et al., 2016 unless their data do not agree with those reported in the related manuscript by the authors.

We have now improved our references to the recent articles, with regard to the histone binding site (e.g. subsection “CAF-1 has one binding site for a H3-H4 dimer”, first paragraph and subsection “H3-H4 binding activates DNA binding by the Cac1 winged helix domain (WHD)”, second paragraph). These articles do not identify a role for Cac2 in histone binding, which we have demonstrated in our work in the related manuscript. We have also clarified this point in these statements.

Results: "it appears that the primary role of CAF-1 is to promote the formation of an ordered (H3-H4)_2_-DNA complex, the tetrasome." This statement corroborates previous work that should be referenced.

We agree and we have now added a reference to this statement: Smith and Stillman, 1991.

c) The authors need to do a better job of explaining/synthesizing earlier results/conclusions from others (e.g. eLife paper from Liu et al. 2016). Likewise, the conclusions of this paper seem to contradict those of Winkler et al. 2012 from the same laboratory. Yet this paper is not cited in the places where the authors refer to previous studies showing CAF-1 promoting H3-H4 tetramer formation (Liu et al. 2012, 2016) (e.g., Introduction, fifth paragraph). The authors should clarify how the new work differs from previous results both their own and other labs, and discuss potential reasons (different assays, buffers, etc.) why different conclusions have been reached previously. This discussion may become quite technical, but it is essential to clarify the advances made here and distinguish them from previous studies. The current discussion misses the opportunity to clarify how the authors think previous experiments 'got it wrong' on issues of stoichiometry and/or to synthesize the findings coming from other studies.

We have now expanded the Discussion to address these comments. We want to make several additional points regarding the apparent discrepancies with the literature, which pertain to the stoichiometry of the CAF-1-histone interaction:

1) We do not cite the earlier publication from our lab (Winkler et al., 2012), because in the course of this work we have identified reproducibility issues with some of the older data from our lab. We are in correspondence with the journal to rectify these issues in the appropriate manner.

2) Consequently, the only published work that shows biochemical evidence that CAF-1 binds a (H3- H4)_2_ tetramer is based on a FRET assay developed by the Churchill lab (Liu et al. 2012 and 2016), without directly measuring the stoichiometry of the CAF-1•H3-H4 complex. We have not been able to set up a similar FRET assay in a reproducible manner. In our manuscript, we use a variety of techniques to directly probe the stoichiometry of the CAF-1•H3-H4 complex. Some are based on FRET (job plot), but others (analytical ultracentrifugation, HX-MS and SEC-MALS) are label-free approaches.

3) All of these independent approaches documented in the current manuscript support the hypothesis that one CAF-1 complex binds a H3-H4 dimer, which is able to tetramerize with another H3-H4. We should point out that this conclusion is in agreement with the experimental observations in the Liu papers, yet is in contradiction with their interpretation.

4) We would like to point out that in the Liu et al. 2016 paper, the HX-MS results on the histone proteins are omitted. Nonetheless, from the online “author response” letter to the reviewers on the *eLife* website, we deduced that their conclusions from those experiments were not supportive of binding of a (H3-H4)_2_ to a CAF-1 complex, but rather in line with our findings.

We copy the available letter below:

“6) […] Such a scheme could be helpful in rationalizing why there is no change at the H3/H3' 4- helix bundle; at first glance, one would think that these regions should gain protection going from a dimer to a tetramer (based on published HXMS results). It appears there are insufficient (any?) peptides that could be used to measure this histone interaction – can the authors comment why this might be?

Please see point 7 below. 7) […]

We used a slightly different buffer with a different pH in order to stabilize the H3/H4 alone sample for the HX experiment. Therefore, the results between the H3/H4 and CAF-1-H3/H4 samples should not be compared, as the buffer differences alone may explain the differences in HX. Accordingly, we have removed the histone dataset from the manuscript. The HX data indeed had poor coverage, especially around the histone N-terminal tails and α3 of H3; however, the CX data has much better coverage and is sufficient to support our conclusion that CAF-1 interacts with both the N-terminal tails and α helical cores of H3 and H4. Removing these data do not significantly change any conclusions of this study.”

To our knowledge, our finding that CAF-1 binds a H3-H4 dimer that is capable of forming a (H3-H4)_2_ tetramer, either with or without a second CAF-1 molecule, is consistent with the published data, with the exception of the Winkler paper from our lab, which is in the process of being corrected.

We have included a streamlined version of these observations in the Discussion.

*d) The highly speculative discussion of the authors' ideas on how stoichiometry might yield inheritance of histones at replication should be shortened/eliminated as it is not clear that the current studies provide new insights into this process. Indeed, it is not clear that CAF-1 functions in this process at all (CAF-1 may only function during new histone deposition).*

We have now shortened this part.

2) The authors use HX-MS experiments to probe changes in CAF-1 conformation upon H3-H4 binding. Similar experiments were carried out by Liu et al. 2016. The authors should compare the data sets and highlight potential differences or, if there is not substantial new information, remove the figure.

We have now improved the representation of our HX-MS data, in light of the Liu 2016 paper (now in subsection “H3-H4 binding activates DNA binding by the Cac1 winged helix domain (WHD)”, first paragraph). We decided to maintain our HX-MS data in the manuscript for the following reasons:

1) We have used a different Cac1 construct (tCac1, instead of FL Cac1 that is used in the Liu 2016 paper) and the data included here demonstrate that tCac1 behaves similarly to FL-Cac1 used in Liu et al.

2) We include HX-MS results from two different H3-H4 isoforms (WT and DM), which show the same conformational effects on CAF-1. In the Liu 2016 paper, the authors have not included the deuterium uptake results for the histones, which makes this part of our data novel and important for the complete understanding of the complexes.

3) Compared to Liu et al., we have measured additional time points (1-10-30-60 minutes).

4) We have identified more peptides overall than in the previous study, which provides a higher resolution look at the HX changes in CAF-1.

5) The results are largely confirmatory, but also serve two important purposes – that is, 1) to build on the Liu et al. data set by increasing the amount of peptide coverage and HX data collected and, 2) to provide confidence in the HX-MS patterns presented in both manuscripts.

3) The model presented in Figure 6 predicts that the WHD interacts with the acidic region. Can the authors confirm this with direct binding assays or is there already evidence for this in the recent crosslinking studies of CAF-1 (Liu et al., 2016; Kim et al., 2016)?

In the original paper, we presented support for this hypothesis by performing a competition experiment (Figure 3). Indeed, in the Liu et al. 2016 paper, they report one EDC cross-link between the Cac1 WHD (aa 583) and a residue adjacent to the acidic region in the Cac2-binding region (aa 460, no cross-links are reported for the acidic regions in their MS experiments) (Figure 3—figure supplement 1 and [Supplementary-material SD4-data] in Liu et al. 2016). Strikingly, this cross-link is not detected when H3-H4 are present (Figure 3—figure supplement 1 in Liu et al. 2016), in line with our data that histone binding releases the WHD from the intramolecular interaction.

We have now added in solution cross-linking experiments that further validate that in presence of the histone binding module, which contains the Cac1 acidic region and Cac2, we can see the formation of a higher molecular weight complex, which is not formed when we use only the Cac2 subunit, as control (new Figure 3—figure supplement 1 and subsection “H3-H4 binding activates DNA binding by the Cac1 winged helix domain (WHD)”, second paragraph), confirming our interpretation that an interaction between the WHD and the acidic region can occur.

*4) As mentioned by the authors, crosslinking experiments showed contacts between the WHD and histones. This seems at odds with the model predicted in this paper. Could it be that the WHD aids histone binding for non-chromatin bound CAF-1, and* in vivo *only switches to bind DNA once the N-terminal region has recruited CAF-1 to PCNA and new DNA? The authors should consider this possibility.*

If we understand this question correctly, the reviewer is proposing a model where PCNA binding may affect the binding “partners” of the Cac1 WHD, which could switch from histone to DNA binding.

It is possible that PCNA may induce structural changes that affect CAF-1 architecture and we included this idea in the fourth paragraph of the Discussion. However, we have no evidence that the WHD contributes to histone binding:

1) Liu et al. (in their Figure 4 in the 2016 paper) report a marginal binding affinity between the WHD and histones, significantly lower than what is measured for the CAF-1 complex or the histone binding module.

2) The cross-linking data may result from the physical proximity of the WHD to the histone binding site that we identify here (Figure 3), rather than a direct interaction, as described in our Discussion (third paragraph).

3) We show that a WHD deletion in CAF-1 does not affect the binding affinity and stoichiometry of CAF-1 to H3-H4 (Figure 3—figure supplement 1).

We have therefore no data to support the idea that the WHD may at any time engage in direct histone binding, so we decided not to include this concept in the manuscript.